# UNiMMVSR: A UNIFIED CASCADED FRAMEWORK WITH MULTI-MODAL VIDEO SUPER-RESOLUTION

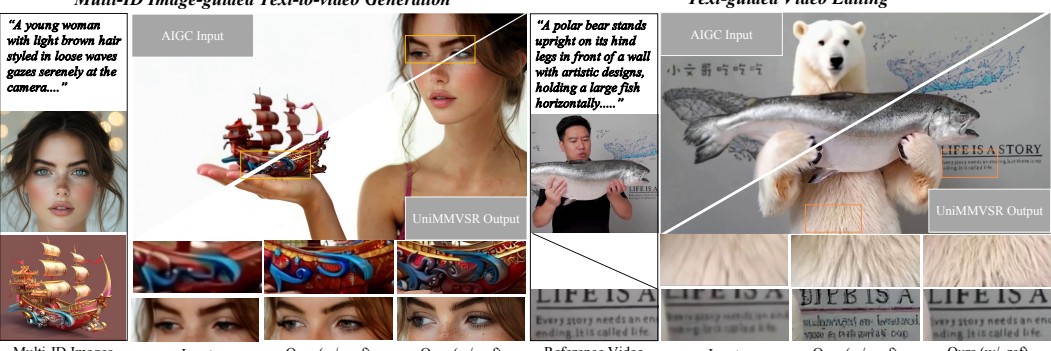

Figure 1: **UniMMVSR is a unified framework that supports video super-resolution with multi-modal input conditions.** By cooperating with the low-resolution multi-modal generative model, the proposed cascaded framework can effectively extend the controllable video generation to ultra-high-resolution (e.g., 4K) with high visual quality and subject consistency.

## ABSTRACT

Cascaded generation framework has emerged as a promising technique for decoupling the computational burden associated with generating high-resolution videos using large foundation models. Existing studies, however, are largely confined to text-to-video tasks and fail to leverage additional generative conditions beyond text, which are crucial for ensuring fidelity in multi-modal video generation. We address this limitation by presenting UniMMVSR, the first unified generative video super-resolution framework to incorporate hybrid-modal conditions, including text, images, and videos. We conduct a comprehensive exploration of condition injection strategies, training schemes, and data mixture techniques within a latent video diffusion model. A key challenge was designing distinct data construction and condition utilization methods to enable the model to precisely utilize all condition types, given their varied correlations with the target video. Our experiments demonstrate that UniMMVSR significantly outperforms existing methods, producing videos with superior detail and a higher degree of conformity to multimodal conditions. We also validate the feasibility of combining UniMMVSR with a base model to achieve multi-modal guided generation of 4K videos—a feat previously unattainable with existing techniques.

## 1 INTRODUCTION

Video generation foundation models (Seed, 2025; Wan et al., 2025; Hu et al., 2025) have made remarkable progress in synthesizing realistic videos, largely due to the scaling law of diffusion transformer architectures (Peebles & Xie, 2023). Unfortunately, expanding model capacity typically incurs a significant computational burden, a challenge particularly pronounced for high-resolution video generation (e.g., 2K, 4K, 8K), a growing trend in future applications. To resolve this dilemma, a stage-wise cascading paradigm, where a large-capacity base model generates a low-resolution video and subsequent lightweight super-resolution models synthesize the fine details, has emerged

as a promising solution. Anyhow, existing research (Ho et al., 2023; Zhang et al., 2025a; Seed, 2025)is limited to text-to-video generation task. A significant gap remains in understanding how super-resolution models can effectively use hybrid conditions—a crucial capability for maintaining generative fidelity in videos produced by multi-modal base models.

In this paper, we present the first unified latent diffusion framework for multi-modal video super-resolution, dubbed UniMMVSR. We focus on three common video generation tasks: text-to-video generation, multi-ID image-guided text-to-video generation, and text-guided video editing. For these tasks, our super-resolution model uses not only the low-resolution video but also text, ID images, and other videos as conditions. The main challenge is to integrate these diverse conditions into a single framework and to modulate these reference information in a compatible manner. This is crucial to ensure the model to use all conditions effectively, allowing it to generate vivid details that conform to the multi-modal guidance.

To achieve this, we conducted a thorough study on multi-modal condition injection, with a special focus on incorporating multiple ID images and reference videos. Our comparative analysis shows that token concatenation performs best among the baselines. Recognizing that the low-resolution video from a base model might not perfectly align with the multi-modal conditions, we improved the robustness of UniMMVSR in two ways: (i) We assign independent position embedding for condition tokens that are distinct from the noisy target video tokens. This encourages the model to use the references based on context and correlation even though their contents are pixel-aligned. (ii) We developed a custom training data pipeline that simulates the generation characteristics of base models using the SDEdit technique (Meng et al., 2021).

Our experiments prove that UniMMVSR is superior to existing baselines, especially in its visual fidelity to multi-modal references. Our ablation studies further validate the effectiveness of our key designs, offering a clear view of the advantages of our method. We also show the benefits of our unified training framework: high-quality training data can transfer across sub-tasks, which reduces the burden of collecting high-quality data for complex-modal tasks.

Our contributions are summarized as follows:

- We introduce UniMMVSR, the first multi-modal guided generative video super-resolution model built on a cascaded framework. Our model synthesizes vivid details while maintaining high fidelity to conditional references.

- We developed a unique SDEdit-based degradation pipeline to create synthetic training data for multi-modal video super-resolution. It enhances the model's robustness to discrepancies between low-resolution video inputs and multi-modal conditions.

- Our UniMMVSR framework demonstrates the ability to leverage high-quality training data across multiple tasks and can easily scale to ultra-high-resolution generation (e.g., 4K) with efficient computational overhead.

## 2 RELATED WORKS

### 2.1 MULTI-MODAL VIDEO GENERATION

With the advancement of video generative model, recent work has increasingly focused on enhancing the controllability of generated videos. (Huang et al., 2025; Chen et al., 2025; Yuan et al., 2025; He et al., 2024b; Hu, 2024; Lei et al., 2025; Ma et al., 2024; Wei et al., 2024; Zhang et al., 2025b) introduced reference images to improve subject consistency in the output video, while methods such as (Chen et al., 2024; Tu et al., 2025; Mou et al., 2024; Ye et al., 2025; Liew et al., 2023) achieved mask-based or instruction-based video editing by incorporating referenced videos. Although these approaches demonstrate promising results for specific controllable tasks, they fail to generalize across diverse tasks, hindering the broader application of controllable video generation.

To establish a unified framework for controllable generation tasks, previous work (Ding et al., 2022; Ju et al., 2023) introduced multiple adapter modules to independently incorporate different reference conditions. This approach yielded poor performance while resulting in significant waste of model parameters. Consequently, recent methods such as FullDiT (Ju et al., 2025; Tan et al., 2025) leverage

in-context condition mechanisms to flexibly combine multi-modal input condition signals through self-attention module, achieving multi-task controllable video generation in a unified framework.

However, the computational complexity of the self-attention module increases quadratically with the number of tokens, hindering the scalability of such methods to more tasks and higher resolutions. While FullDiT2 (He et al., 2025) optimizes computational overhead for reference conditions through kv cache, block skipping, and token selection techniques, it remains inapplicable to unified frameworks or high-resolution scenarios. To address this limitation, we propose the first unified cascaded framework for high-resolution multi-modal video generation. This approach effectively achieves controllable high-resolution video generation while faithfully preserving multiple input conditions.

## 2.2 VIDEO SUPER-RESOLUTION

Most previous works (Chan et al., 2022b; Cao et al., 2021; Chan et al., 2021; 2022a) primarily focus on synthetic or real-world data by designing compositional synthetic degradation pipelines to model the degraded videos. With the widespread applications of video generation models, later approaches shift towards AI-generated data. Due to limited generative capabilities, previous methods tend to generate over-smooth results. Motivated by recent advances in diffusion models, several diffusion-based video super-resolution (VSR) methods (Wang et al., 2023b; Zhou et al., 2024; Yang et al., 2024; He et al., 2024a; Li et al., 2025; Wang et al., 2025b;a) have been proposed, which show impressive performance and generate realistic details.

However, limited by recent video super-resolution framework, existing models can only take text prompt and input video as conditions, which hinders their applications towards controllable video generation tasks. Although producing fine details, due to the randomness of diffusion sampling process, it inevitably reduces the fidelity of the input video to multi-modal references, which further diminishes the controllability of the generated results. In this paper, for the first time, we design a generative video super-resolution framework that unifies the input of hybrid-modal conditions, which improves the visual quality of the input video while ensuring its fidelity to multi-modal conditions.

## 3 METHOD

We aim to achieve generative video super-resolution for AI-generated videos under hybrid multi-modal conditions, which synthesizes rich, vivid details and maintains high fidelity to various conditional inputs. It works in the scenario that multi-modal base models first generates a low-resolution video, which our UniMMVSR model then upscales using the original high-resolution conditions if they're available. The overview flowchart is depicted in Fig. 2. Specifically, we focus on three common video generation tasks: Text-to-video, Text-to-video guided by multiple ID images, and Text-guided video editing. To accomplish this, our super-resolution model incorporates diverse inputs—including low-resolution video, text, multiple ID images, and reference videos—in a compatible manner. We have tackled this challenge by exploring a unified condition injection mechanism, a custom training data pipeline, and a tailored training strategy to ensure the model effectively utilizes all multi-modal conditions.

### 3.1 PRELIMINARIES

Our model is built upon a pretrained large-scale text-to-video latent diffusion model. It first pretrains an autoencoder that converts a video $x$ into a low-dimensional latent $z$ with an encoder $\mathcal{E}$ and reconstructs it with a decoder $\mathcal{D}$. The core of this framework is a conditional diffusion transformer that operates in the compressed latent space. Detailed architecture is presented in Supp. A.1.

During training, given a LR-HR paired data $(z_{HR}, z_{LR})$ and multiple conditions $C$, isotropic gaussian noise is added to generate corresponding noise latent $z_t = (1 - t)z_{HR} + t\epsilon$, where $\epsilon \in \mathcal{N}(0, I)$. With the formation of flow matching, it trains a network $\mu_\theta(z_t, t, C)$ to predict the velocity $v = z_{HR} - \epsilon$. Then, the network $\mu_\theta$ is optimized by minimizing the mean squared error loss $\mathcal{L}$ between the ground truth velocity and the model prediction:

$$\mathcal{L} = \mathbb{E}_{z_{HR}, \epsilon, t, C} \| v - \mu_\theta(z_t, t, C) \|. \tag{1}$$

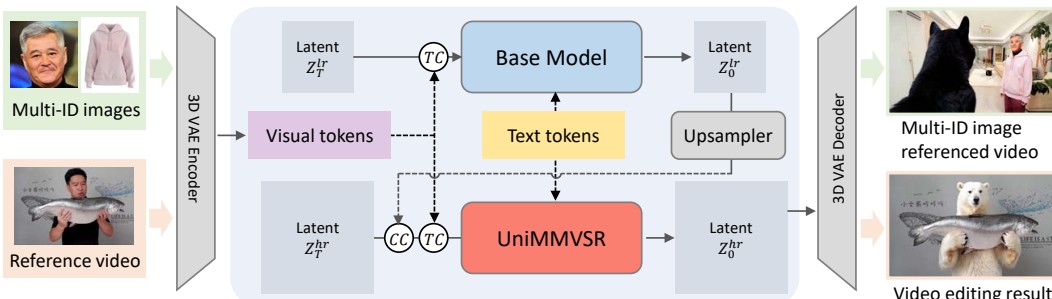

Figure 2: Overview of UniMMVSR in the context of a cascaded generation framework. Upsampler denotes the sequential operations of VAE decoding, upscaling via bilinear interpolation, and VAE encoding. *TC* and *CC* denote token concatenation and channel concatenation respectively. Texts are encoded by text encoder and then injected via cross-attention layers, which are omit for simplicity.

During inference, it first samples noise $\epsilon \in \mathcal{N}(0, I)$, then denoise it by a pre-defined ODE solver with a discrete set of $N$ timesteps to generate clean latent $z_{HR}$. The final output $x_{HR}$ is obtained by projecting $z_{HR}$ to the pixel space using pre-trained decoder $\mathcal{D}$.

## 3.2 UNIFIED CONDITIONING FRAMEWORK

Our UniMMVSR model processes four types of generative conditions: text prompts, multiple ID images, reference videos, and a low-resolution input video. Since UniMMVSR is adapted from a pre-trained text-to-video model, it inherits the original text-prompt conditioning design. We mainly discuss the interaction strategy of conditional visual tokens and noisy target video tokens.

**Low-resolution video via channel concatenation.** In the video super-resolution task, the basic structure of the input video needs to be preserved, while the high-frequency patterns are corrupted by the diffuse process to avoid confusion. We propose to use channel concatenation to incorporate the conditional information since it provides strict tempo-spatial correspondences between the input video and the target high-resolution (HR) video. Note that, most VAE latent spaces do not support simple interpolation, which can cause significant structural distortion. Thus, for the low-resolution (LR) latent generated by the base model, we first decode it to the pixel space, upscale it via bilinear interpolation, and then encode it back into the latent space. This preserves the original information in the LR latent and makes it ready for channel concatenation with the noisy HR latent.

**Visual references via token concatenation.** We refer to multi-ID images and reference videos as visual references. Following the successful practice of recent in-context conditioning methods (Tan et al., 2025; Ju et al., 2025), we integrate the target video tokens with the conditional tokens using token concatenation in the temporal dimension. This strategy makes it feasible to incorporate general conditional modalities into the generation process, as they interact with the target video tokens in each layer's attention modules. To align the spatial size, the referenced images/videos are zero-padded in the pixel space before VAE encoding. Afterwards, in each transformer block, noisy video tokens and visual references are processed in parallel through 2D self-attention, 2D cross-attention, and a feedforward network, which preserves the alignment between the text and video modalities. For the 3D self-attention module, all tokens are treated as a single unified sequence and processed together, which ensures a bidirectional flow of information between the noisy and reference tokens. Finally, all reference tokens are truncated from the transformer output to match the input shape.

**Separated conditional RoPE.** Before concatenation with the target video tokens, these conditional tokens are assigned with position encoding, where we adopt Rotary Position Embedding (RoPE). For multi-ID images, it is natural to assign an individual range of RoPE that are distinct from that of target video tokens, since no direct spatial correspondence exists between them. For reference video, since the LR video generated by our base model is not perfectly pixel-aligned with it, we also assign a separate range of RoPE for thest conditional tokens, so as to encourage our model to utilize it based on context and correlation rather than direct copy-and-paste. Specifically,

we assign indices 0 to $f-1$ for noisy token, $n_i$ to $n_i + k_i$ for $i$-th reference tokens, where $f$ denotes the number of frames, $n_i$ and $k_i$ denotes the start index and length of $i$-th reference token.

### 3.3 DEGRADATION PIPELINE

Multi-modal conditional video generation model requires not only the vividness of generated content but also the conformity to provided conditions. Accordingly, our UniMMVSR model also needs to achieve high-quality details and maintaining fidelity to the multi-modal conditions. Given a high-resolution video, we need to design degradation pipeline to process it into a low-resolution video that should share the characteristics of the base models' output. Specifically, the degradation of multi-modal base model can be categoried into two scenarios: (i) At low resolution, high-frequency details in training data are lost due to resize operations. The base model can only generate basic structures that align with the semantic content of the text prompt, which lacks of fine details and textures. (ii) For some challenging cases, the low-resolution output maintains a low fidelity to the visual references due to the sub-optimal controllability of the base model to harmonize text prompt and visual references. Therefore, the identity in LR video typically exhibits distortion in local structure and has a low visual quality.

To tackle these two scenarios, a custom degradation pipeline need to be designed to simulate the artifacts and distortion generated by the base model. However, traditional degradation pipelines constructed solely based on synthetic degradation factors (such as noise, blur, video compression, etc.) cannot be fully adapted to these scenarios since the resulting local structure of the LR video is strictly aligned with the HR video, failing to simulate the insufficient reference response in low-resolution output. To simulate the second degradation scenario, we recognized that **it is equivalent to the results obtained by base model using only text condition**. Therefore, based on the sdedit method (Meng et al., 2021), we constructed compatible high-frequency degradation features using inference result from the text-to-video base model, termed **SDEdit Degradation**.

Specifically, we downsample HR video to a resolution directly achievable by the text-to-video base model. The resized video $x$ is encoded into latent space via a pre-trained 3D VAE encoder, and noise is added by the forward process of the diffusion model for $k$ steps, where the step value $k$ is randomly sampled from $[K_1, K_2]$ and $K_2$ denotes the maximum threshold to retain the main structure of the input video. Subsequently, we perform $k$ steps of denoising process on the noisy latent using the base model, decoding the result via the 3D VAE decoder to obtain the LR video. After sdedit degradation, we apply synthetic degradation factors to the output $x'$ to construct the final LR video. The degradation pipeline and samples are presented in Supp. A.4.

### 3.4 TRAINING STRATEGY

**Training Order.** To train a unified model of hybrid conditions, the training order of subtasks is essential due to the varied difficulty. Instead of generating by text prompt only, multi-ID image-guided text-to-video generation and text-guided video editing tasks tend to synthesize high-fidelity textures and details by utilizing visual conditions and text prompt together, thus resulting in faster convergence speed than text-to-video generation task as shown in Fig. 12. Thus, we perform a difficult-to-easy training strategy, aiming to learn difficult task first, then effectively adapt to easier tasks.

Starting from a pre-trained text-to-video (T2V) model weight, we first train 21-frames text-to-video generation task independently in the first stage. In the second stage, we train 21-frames text-to-video generation and multi-ID image-guided text-to-video generation tasks together with a probability $0.6 : 0.4$, aiming to retain the ability to generate high-definition details from text. Next, all tasks are trained at 21 frames together with a probability $0.5 : 0.3 : 0.2$. Finally, we extend the frame length to 77 (5 seconds) while keeping the probability unchanged.

**Reference Augmentation.** For multi-ID image-guided text-to-video generation task, most testing scenarios include cross-pair data, where the perspective, orientation, and position of the low-resolution output and ID images exhibit greater discrepancies than training datasets. For text-guided video editing task, although the HR video and reference video are strictly pixel-aligned for non-editing area during training, the low-resolution output exhibits a certain degree of error compared with reference video. Since we only construct synthetic datasets for these two tasks (stated in

Table 1: Quantitative Evaluation of UniMMVSR on all three tasks. **Bold** and underlined indicate the best and second-best results, respectively. ↑ indicates higher is better; ↓ indicates lower is better.

| Method | Visual Quality | | | | Subject Consistency | | Video Alignment | | |
|---|---|---|---|---|---|---|---|---|---|
| | MUSIQ↑ | CLIP-IQA↑ | QAlign↑ | DOVER↑ | CLIP-I↑ | DINO-I↑ | PSNR↑ | SSIM↑ | LPIPS↓ |
| **Text-to-video Generation** | | | | | | | | | |
| Base 512×512 | 30.996 | 0.246 | 3.741 | 0.594 | - | - | - | - | - |
| Base 1080P | 46.645 | 0.306 | 4.246 | 0.749 | - | - | - | - | - |
| VEnhancer | **57.171** | 0.367 | 4.214 | 0.733 | - | - | - | - | - |
| STAR | 56.904 | 0.369 | 4.435 | 0.769 | - | - | - | - | - |
| SeedVR | 55.596 | **0.379** | 4.396 | 0.778 | - | - | - | - | - |
| Ours (single) | 56.146 | 0.366 | **4.535** | 0.771 | - | - | - | - | - |
| **Ours (unified)** | 56.418 | 0.371 | 4.500 | **0.778** | - | - | - | - | - |
| **Text-guided Video Editing** | | | | | | | | | |
| Base 512×512 | 35.073 | 0.234 | 3.615 | 0.400 | - | - | 30.191 | 0.699 | 0.364 |
| Base 1080P | 53.616 | 0.383 | 4.247 | 0.634 | - | - | 29.383 | 0.582 | 0.358 |
| Ref Video | 54.249 | 0.365 | 4.131 | 0.571 | - | - | - | - | - |
| VEnhancer | 57.036 | 0.380 | 4.013 | 0.590 | - | - | 28.417 | 0.571 | 0.489 |
| STAR | 56.802 | 0.397 | 4.264 | 0.608 | - | - | 29.421 | 0.631 | 0.397 |
| SeedVR | 57.820 | 0.370 | 4.183 | 0.635 | - | - | 29.535 | 0.597 | 0.413 |
| Ours (no ref) | **59.119** | **0.399** | 4.289 | **0.648** | - | - | 29.615 | 0.581 | 0.429 |
| Ours (single) | 53.388 | 0.348 | 4.302 | 0.597 | - | - | **31.905** | **0.723** | **0.276** |
| **Ours (unified)** | 53.245 | 0.344 | **4.305** | 0.597 | - | - | 31.556 | 0.713 | 0.282 |
| **Multi-ID Image-guided Text-to-video Generation** | | | | | | | | | |
| Base 512×512 | 29.314 | 0.255 | 3.149 | 0.433 | 0.692 | 0.538 | - | - | - |
| Base 1080P | 46.780 | 0.345 | 4.092 | 0.662 | 0.691 | 0.507 | - | - | - |
| VEnhancer | 60.656 | **0.469** | 4.149 | 0.707 | 0.671 | 0.533 | - | - | - |
| STAR | 58.810 | 0.449 | 4.282 | **0.763** | 0.696 | 0.546 | - | - | - |
| SeedVR | 54.491 | 0.419 | 3.960 | 0.708 | 0.693 | 0.543 | - | - | - |
| Ours (no ref) | 60.947 | 0.445 | 4.385 | 0.742 | 0.693 | 0.543 | - | - | - |
| Ours (single) | 61.357 | 0.446 | 4.414 | 0.743 | **0.728** | **0.566** | - | - | - |
| **Ours (unified)** | **62.248** | 0.465 | **4.428** | 0.745 | 0.726 | **0.566** | - | - | - |

Sec. A.2), directly utilizing synthetic reference conditions leads to train-test gap, thereby compromising the performance on the test set. To mitigate this issue, we design customized reference augmentation technique to narrow this gap. Specifically, we apply several image-related transformations to simulate the cross-pair test scenarios of multi-ID image-guided text-to-video generation task. For text-guided video editing task, we randomly shift the start frame of reference video, aiming to encourage the model to learn more robust context-injection mechanism rather than directly copying the pixels from reference video.

## 4 EXPERIMENTS

### 4.1 EXPERIMENTAL SETTINGS

#### 4.1.1 IMPLEMENTATION DETAILS

Our model is trained on NVIDIA H800 GPUs with a total batch size of 32. The training phase contains 3 stages, and it takes around 1 day per stage. AdamW (Loshchilov, 2017) is used as the optimizer with a learning rate of $10^{-4}$. The text prompt is randomly replaced by a null prompt with $10\%$ probability. To enhance the robustness of our model to different degradation scenarios, we utilize the noise augmentation technique by injecting noise into the input latent using a diffuse process. The noise timestep is randomly sampled from 200 to 600 to preserve the main structure. We have additionally encoded noise timestep as a micro condition for the model. We use a pretrained T2V model to provide initialization weight. During inference, we perform 50 PNDM (Liu et al., 2022) sampling steps with independent classifier-free guidance as stated in Sec. A.3. The guidance scale $s_{txt}$ and $s_{ref}$ are set to 3.0 and 1.0 respectively, with reference guidance threshold $N_{ref} = 15$. We have also used timestep shift (Esser et al., 2024) with shift value 1.0.

#### 4.1.2 TESTING SETTINGS

**Baseline Methods.** To evaluate the effect of our cascaded framework, we compare with the end-to-end results of our base model (both 512×512 and 1080P). Since there is limited work on multi-

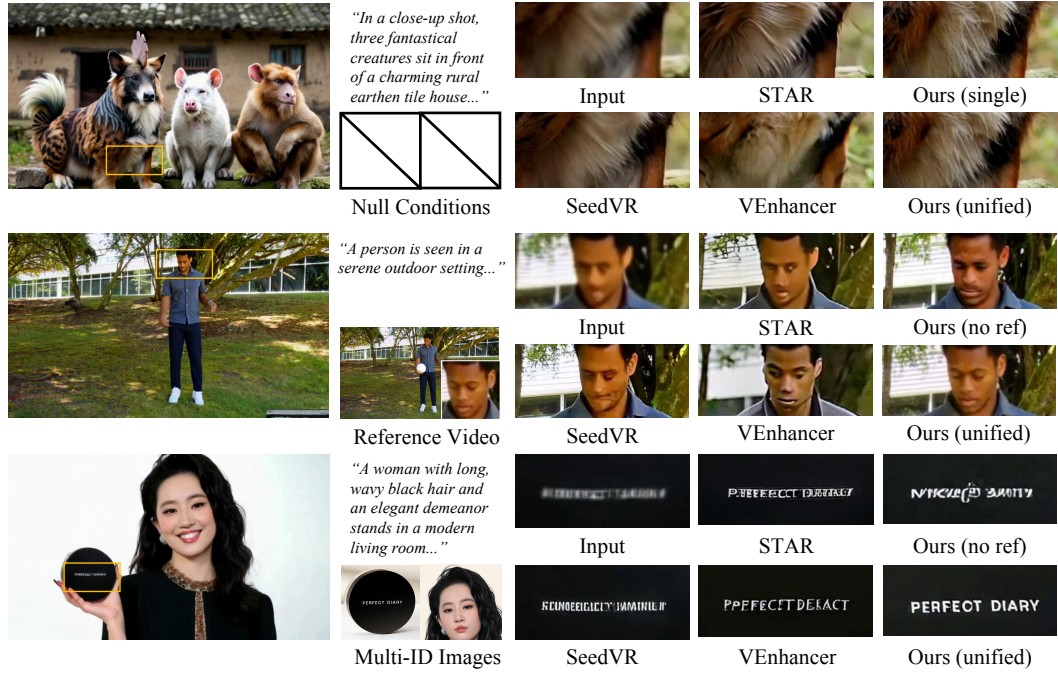

Figure 3: Qualitative comparisons on *text-to-video generation*, *text-guided video editing* and *multi-ID image-guided text-to-video generation* tasks from top to bottom. (**Zoom-in for best view**)

modal VSR tasks, we also compare UniMMVSR with state-of-the-art VSR methods VEnhancer (He et al., 2024a), STAR (Xie et al., 2025) and SeedVR (Wang et al., 2025b;a).

**Evaluation Metrics.** For visual quality, we conduct our evaluation of commonly used visual quality metrics MUSIQ (Ke et al., 2021), CLIP-IQA (Wang et al., 2023a), Q-Align (Wu et al., 2023b) and DOVER (Wu et al., 2023a). For multi-ID image-guided text-to-video generation task, we utilize DINO-I (Caron et al., 2021) and CLIP-I (Radford et al., 2021) to assess the fidelity to multiple ID images. We additionally use PSNR, SSIM and LPIPS (Zhang et al., 2018) to evaluate alignment of non-editing area with the reference video for text-guided video editing task.

## 4.2 QUANTITATIVE COMPARISON

Quantitative comparisons are shown in Tab. 1. Results show that although the UniMMVSR integrates multiple conditions, it still achieves state-of-the-art performance on controlling metrics compared with base model, previous VSR methods and our method without reference conditions, thereby validating the effectiveness of our method. For visual quality, UniMMVSR obtains the best QAlign&DOVER scores on text-to-video generation task and MUSIQ&QAlign scores on multi-ID image-guided text-to-video generation task, indicating its high perceptual quality. For text-guided video editing task, it is worth noting that our method maintains high pixel-level fidelity and structural similarity to the reference video for non-editing area, thus achieving similar metric values to the reference video. Even so, our approach remains competitive, achieving the best QAlign score. Furthermore, on multi-ID image-guided text-to-video generation task, our unified model exhibits high perceptual quality than our single-task model, indicating that complex-modal tasks can benefit from high-quality text-to-video data, effectively lowers the barrier to collect high-quality reference-video paired data. More comprehensive results can be seen in Supp. A.5.2.

## 4.3 QUALITATIVE COMPARISON

Fig. 3 shows visual results on all three tasks. For text-to-video generation, both our single-task and unified model effectively remove existing degradation patterns and generate fine details like the dog's fur, while other approaches produce blurred details. For text-guided video editing and multi-ID image-guided text-to-video generation, UniMMVSR successfully leverages ID images and

Table 2: Ablation Study of UniMMVSR components on the multi-ID image-guided text-to-video generation Task. We analyze the impact of each component by visual quality and controlling metrics.

| Ablation | Variant | MUSIQ↑ | CLIP-IQA↑ | QAlign↑ | DOVER↑ | CLIP-I↑ | DINO-I↑ |
|---|---|---|---|---|---|---|---|
| Ours | - | 62.248 | 0.465 | 4.428 | 0.745 | 0.726 | 0.566 |
| Architecture Design | full channel-concat | 61.146 | 0.461 | 4.399 | 0.748 | 0.690 | 0.546 |
| | full token-concat | 61.974 | 0.464 | 4.442 | 0.739 | 0.728 | 0.565 |
| Degradation Effect | synthetic degradation only | 62.541 | 0.458 | 4.408 | 0.749 | 0.717 | 0.561 |
| | sdedit degradation only | 59.697 | 0.437 | 4.357 | 0.726 | 0.730 | 0.564 |
| Training Order | full training | 62.199 | 0.460 | 4.322 | 0.745 | 0.716 | 0.553 |
| | easy-to-difficult | 61.706 | 0.445 | 4.326 | 0.736 | 0.717 | 0.556 |

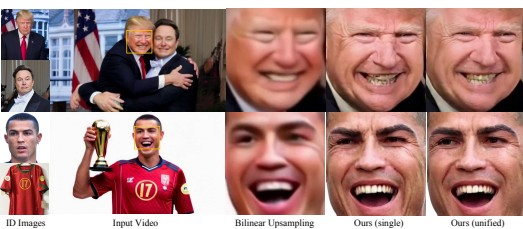 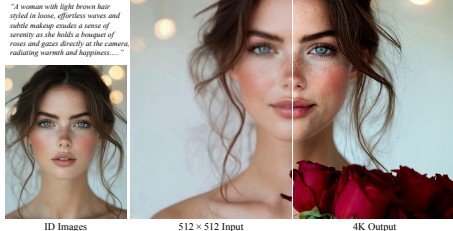

Figure 4: Visual Comparisons of single-task and unified model. **Zoom-in for best view.**

Figure 5: Qualitative results of 4K multi-ID image-guided text-to-video generation.

reference videos to generate high-fidelity textures and details, such as the facial structure of the man and the words "perfect diary" on the box. More results can be found in Supp. A.5.3.

## 4.4 ABLATION STUDY

Due to space limit, we provide qualitative evaluation in Supp. A.5.4.

**Architecture Design.** In Tab. 2, we compare UniMMVSR with two architecture designs: full channel-concat and full token-concat. The former represents concatenating input video and reference tokens along channel dimension, while the latter represents concatenating along sequence dimension. As can be seen, full channel-concat method results in severe performance degradation in controlling metrics (0.690 vs 0.726 for CLIP-I and 0.546 vs 0.565 for DINO-I), which shows that it faces difficulties in reference injection. For full token-concat, while it achieves comparable performance, it results in nearly $2\times$ computational burden due to the quadratic computation complexity.

**Degradation Effect.** Since our degradation pipeline comprises both synthetic and sdedit degradations, we perform ablation study to investigate the effectiveness of each component. As can be seen in Tab. 2, although only using synthetic degradation obtains similar visual quality metrics, it leads to poorer quality in controlling metrics, which confirms that sdedit degradation successfully simulates the degradation scenario of base model results. Next, to validate the necessity of traditional synthetic degradation pipeline, we use sdedit degradation only to construct LR data. While it achieves comparable controlling metrics, it shows a decline in all visual quality metrics, demonstrating the effectiveness of synthetic degradation in detail synthesis.

**Training Order.** To form a unified model, we have implemented three different training strategies: difficult-to-easy, easy-to-difficult and full training. Specifically, difficult-to-easy means training in the order: text-to-video →multi-ID image-guided→video editing, whereas easy-to-difficult denotes the reverse order. Full training represents training all tasks together from the pre-trained T2V model weight. In Tab. 2, we demonstrate the validness of our proposed training strategy by comparing with other strategies above. By utilizing a difficult-to-easy training order, UniMMVSR successfully adapts to multiple tasks while maintaining the performance on previous tasks.

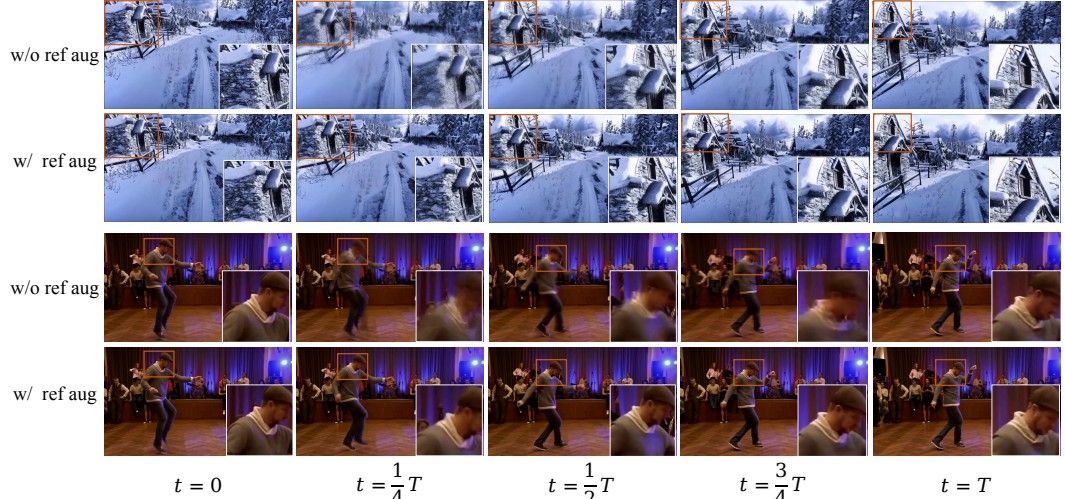

$$t = 0 \qquad t = \frac{1}{4}T \qquad t = \frac{1}{2}T \qquad t = \frac{3}{4}T \qquad t = T$$

Figure 6: Visual Comparisons of reference augmentation. **Zoom-in for best view.**

## 4.5 DISCUSSION

**Importance of Reference Augmentation.** Fig. 6 shows visual comparisons on text-guided video editing task. As can be seen in the second frame in the upper part and the middle three frames in the lower part, training without reference augmentation tend to produce unstable structure, which results in temporal jitter in some frames. After using reference augmentation, UniMMVSR learns to preserve the basic structure of the input video and avoids direct replication of the reference video, which mitigates the conflict of the basic structure between reference video and LR video.

**High-quality Data Transfer across Sub-tasks.** We perform an ablation study by training a single-task model on multi-ID image-guided text-to-video generation task without quality filtering. The model is compared with a unified model mix-trained on a high-quality text-to-video generation dataset. The results are shown in Fig. 4. As can be seen, the unified model generates more natural details such as teeth structure and facial expression, which demonstrates that the high resolution training data can transfer across sub-tasks.

**Resolution Scaling Ability of Cascaded Framework.** Due to the scarcity of ultra-high-resolution (UHR) reference-video paired dataset and quadratic computational complexity, it is difficult to directly train a high-resolution controllable video generative model. By decoupling the process as low-resolution basic structure generation and high-frequency detail synthesis, UniMMVSR successfully generates 4K videos under multi-modal guidance. As shown in Fig. 5 and Supp. A.5.5, our method not only generates vivid details, but also preserves information in reference conditions.

## 5 CONCLUSION

In this paper, we introduce UniMMVSR, the first multi-modal guided generative video super-resolution model built on a cascaded framework. By treating the visual references as a unified sequence and processing them via the 3D self-attention module, UniMMVSR effectively synthesizes vivid details while maintaining high fidelity to conditional references. To enhance the model's robustness to discrepancies between low-resolution video inputs and multi-modal conditions, we develop a unique degradation pipeline based on sdedit method, which simulates the insufficient reference response in low-resolution output. Furthermore, we design a tailored training strategy to form a unified model, and demonstrate that high-quality training data can transfer across sub-tasks, which reduces the burden of collecting high-quality data for complex-modal tasks. The proposed cascaded framework shows its resolution scaling ability, which achieves multi-modal guided 4K video generation for the first time.

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

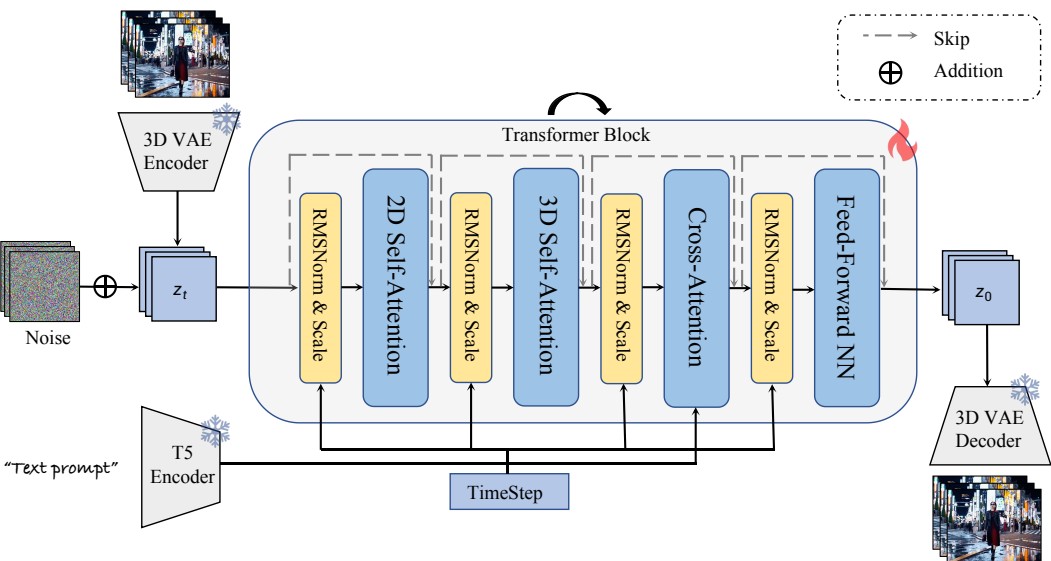

Figure 7: An overview of the architecture of our base model.

## A APPENDIX

The content in the appendix is categorized as follows:

- Base Model.
- Training Datasets.
- Inference Technique.
- SDEdit Degradation.
- More Results.
    - Training Convergence Speed.
    - Quantitative Comparisons.
    - Qualitative Comparisons.
    - Ablation Study.
    - 4K Results.
    - Video Results.
    - Selection of the RoPE Interval.
    - Noise Augmentation Effect.
    - Robustness Against Sub-task Probability.
    - Computational Complexity Analysis.
    - Comparison with a Small Backbone.
    - User Study.
- Details of Test Set.
- Limitations.
- Ethics Statement.
- Reproducibility Statement.

### A.1 BASE MODEL

The architecture of our base model is shown in Fig. 7. Our UniMMVSR is built upon a pre-trained DiT-based video diffusion model, which comprises four main components: spatial self-attention (SSA), spatial cross-attention (SCA), temporal self-attention (TSA) and feed-forward network (FFN). Text prompts, time step and other micro conditions (aspect ratio, FPS, etc) are injected

Table 3: Image-related transforms of reference augmentation. We denote the height and width of the reference image as $h$ and $w$.

| Transform | Probability | Hyperparameters | |
|---|---|---|---|
| | | Type | Sampling Range |
| Brightness | 1.0 | scale | $[0.9, 1.1]$ |
| Horizontal flip | 0.5 | - | - |
| Shearing (x-axis) | 1.0 | value (px) | $[-0.05, 0.05] \times w$ |
| Shearing (y-axis) | 1.0 | value (px) | $[-0.05, 0.05] \times h$ |
| Rotation | 1.0 | value (°) | $[-20, 20]$ |
| Random crop | 0.5 | scale | $[0.67, 1.0]$ |

via the modulation mechanism (Peebles & Xie, 2023). The pre-trained model is trained on 77-frames 512×512 resolution high-quality video data with diverse aspect ratio using NaViT Dehghani et al. (2024). The base model comprises a 3D VAE and a latent DiT backbone, with 1.4B and 10.0B parameters respectively.

## A.2 Training Datasets

**Text-to-video Generation.** We train our model using 840K self-collected high-quality video-text pairs, with each clip processed to 5 seconds and 1080P resolution. The dataset is constructed by applying several IQA/VQA methods (Wu et al., 2023b; Wang et al., 2023a; Ke et al., 2021; Wu et al., 2023a) to filter out low-quality data from 5M raw videos. The text prompts are all captioned using LLAVA captioner (Liu et al., 2024), and encoded by T5 text encoder (Raffel et al., 2020) with no more than 512 tokens.

**Multi-ID Image-guided Text-to-video Generation.** Since portrait-related images dominate the application of multi-ID image-guided text-to-video generation task, we collect around 1.5M videos from open-sourced movies and television series. We then apply the same data filtering as text-to-video generation task to obtain 480K high-quality samples for training. We randomly select a non-overlapping frame from the video clip to extract the reference image. Finally, we apply Mask2former method (Cheng et al., 2022) to identify and extract referenced images. During training, we apply several image-related transforms for reference augmentation. Details are shown in Tab. 3. Since UniMMVSR is primarily designed to extract the high-frequency components from the reference image, we have removed several transformations that would impair the clarity and details of the reference image, such as gaussian blur and downscale.

**Text-guided Video Editing.** Although inpainting-based datasets align better with test scenarios, model-generated reference videos naturally lack high-definition details required by our method. Thus, we follow the local-editing data pipeline (Hu et al., 2024) to preserve the high-frequency information in the non-editing area of the reference video, which results in 450K high-quality samples.

## A.3 Inference Technique

UniMMVSR is trained on video data with either reference conditions or null conditions, and thus it can handle both scenarios. During inference, we apply independent classifier-free guidance (CFG) for each condition as:

$$
\begin{aligned}
\tilde{\epsilon}_\theta(z_t, t, c_{txt}, c_{ref}) = {} & \epsilon_\theta(z_t, t, c_{txt}, c_{ref}) \\
& + s_{txt} \cdot (\epsilon_\theta(z_t, t, c_{txt}, c_{ref}) - \epsilon_\theta(z_t, t, \phi_{txt}, c_{ref})) \\
& + s_{ref} \cdot (\epsilon_\theta(z_t, t, c_{txt}, c_{ref}) - \epsilon_\theta(z_t, t, c_{txt}, \phi_{ref})),
\end{aligned}
\tag{2}
$$

where $c_{txt}$, $c_{ref}$ denote the condition of text prompt and reference, $\phi_{txt}$, $\phi_{ref}$ denote the corresponding null conditions and $s_{txt}$, $s_{ref}$ are the guidance scale. However, we find that simply increasing reference scale $s_{ref}$ leads to over-sharpen results and even generates artifacts. Since our goal is to modify the local structure of the input video based on reference conditions and generate

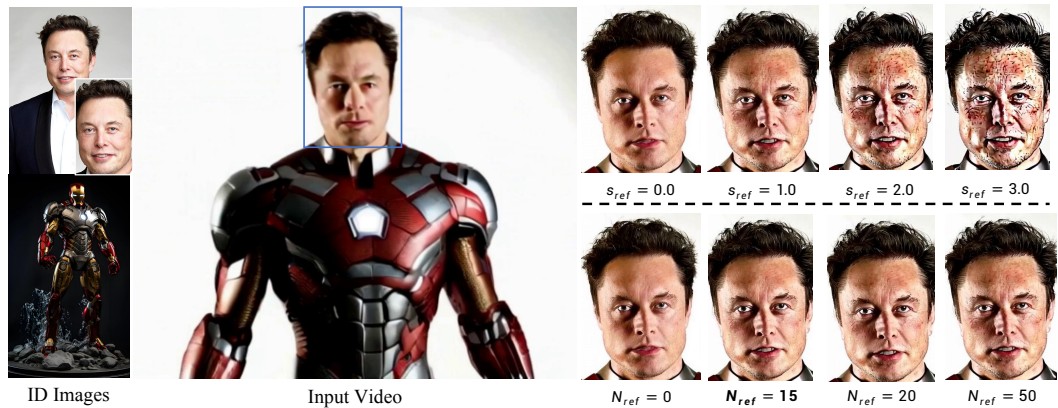

Figure 8: Qualitative comparisons of different inference settings. The text prompt is omitted.

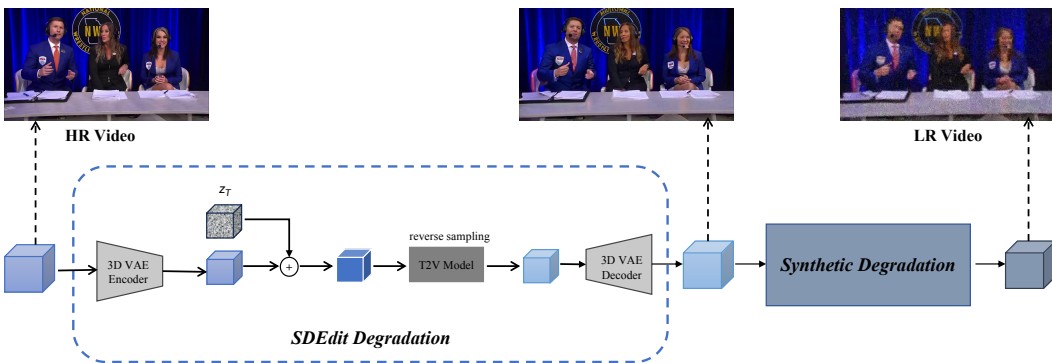

Figure 9: Degradation pipeline for UniMMVSR.

high-frequency details, we introduce reference guidance threshold (RGT) technique to only utilize reference condition for first $N_{ref}$ steps as below:

$$\tilde{s}_{ref} = \begin{cases} s_{ref}, & n < N_{ref} \\ 0, & n \geq N_{ref} \end{cases} \tag{3}$$

We have compared the results of different inference settings in Fig. 8. As shown below, directly increasing reference guidance scale leads to over-sharpen details and even artifacts. By utilizing the proposed RGT technique ($s_{ref} = 1.0 \& N_{ref} = 15$), UniMMVSR enhances the guidance of the reference conditions, further strengthening the generalization on the cross-pair test set.

## A.4 SDEDIT DEGRADATION

The degradation pipeline is shown in Fig. 9. We first perform sdedit degradation to modify the local structure of HR video using the sdedit method by our text-to-video base model. Afterwards, we apply traditional synthetic degradation to introduce high-frequency degradation pattern. Light and heavy sdedit degradation samples are shown in Fig. 10 and 11 respectively.

We conduct additional quantitative experiments to measure the identity similarity using CLIP-I and DINO-I metrics. Specifically, we randomly select 100 videos from training data, and apply Mask2former to extract the reference images from the first frame of the input video. The metrics are computed by the first frame of the input and sdedit-degraded videos. The results are shown in Tab. 4. From these results, we observe that the results of 5 steps maintain a certain level of identity similarity, while the results of 15 steps significantly corrupt the high-frequency identity features. This provides supportive evidence for our assumption that the sdedit degradation effectively mimics the generated video distribution.

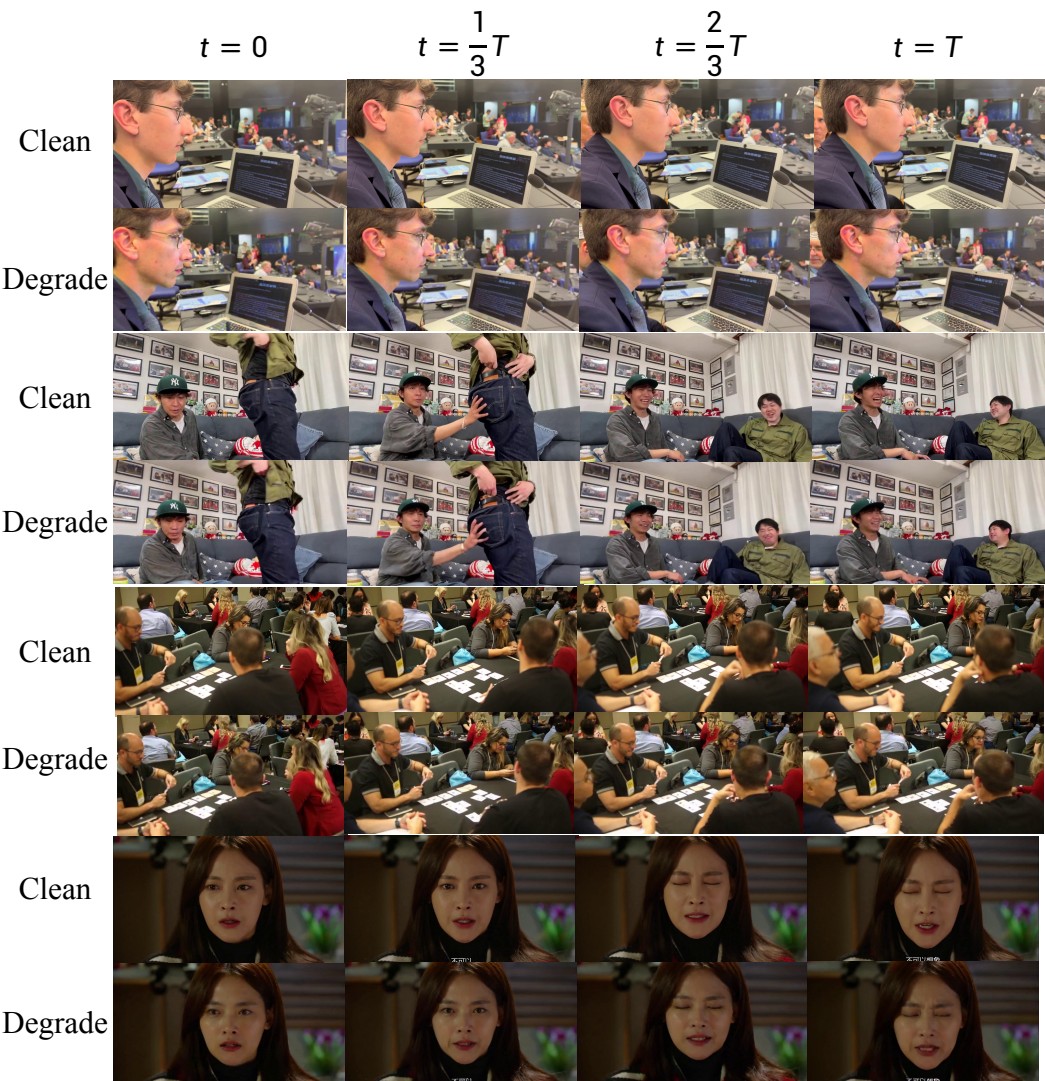

Figure 10: Samples of light sdedit degradation.

Table 4: Quantitative evaluation of the sdedit degradation. The metrics are calculated with the reference images extracted from the first frame of the input video.

| Input | CLIP-I↑ | DINO-I↑ |
|---|---|---|
| Input Video | 0.751 | 0.607 |
| 5-steps Degraded Video | 0.702 | 0.545 |
| 15-steps Degraded Video | 0.671 | 0.522 |

## A.5 MORE RESULTS

### A.5.1 TRAINING CONVERGENCE SPEED

We have shown the training loss curve of single-task model on text-to-video generation, multi-ID image-guided text-to-video generation and text-guided video editing tasks in Fig. 12. As can be seen, text-guided video editing task tends to converge faster at a lower loss value $0.18$, while text-to-video generation task converges slowest, at around $3k$ steps.

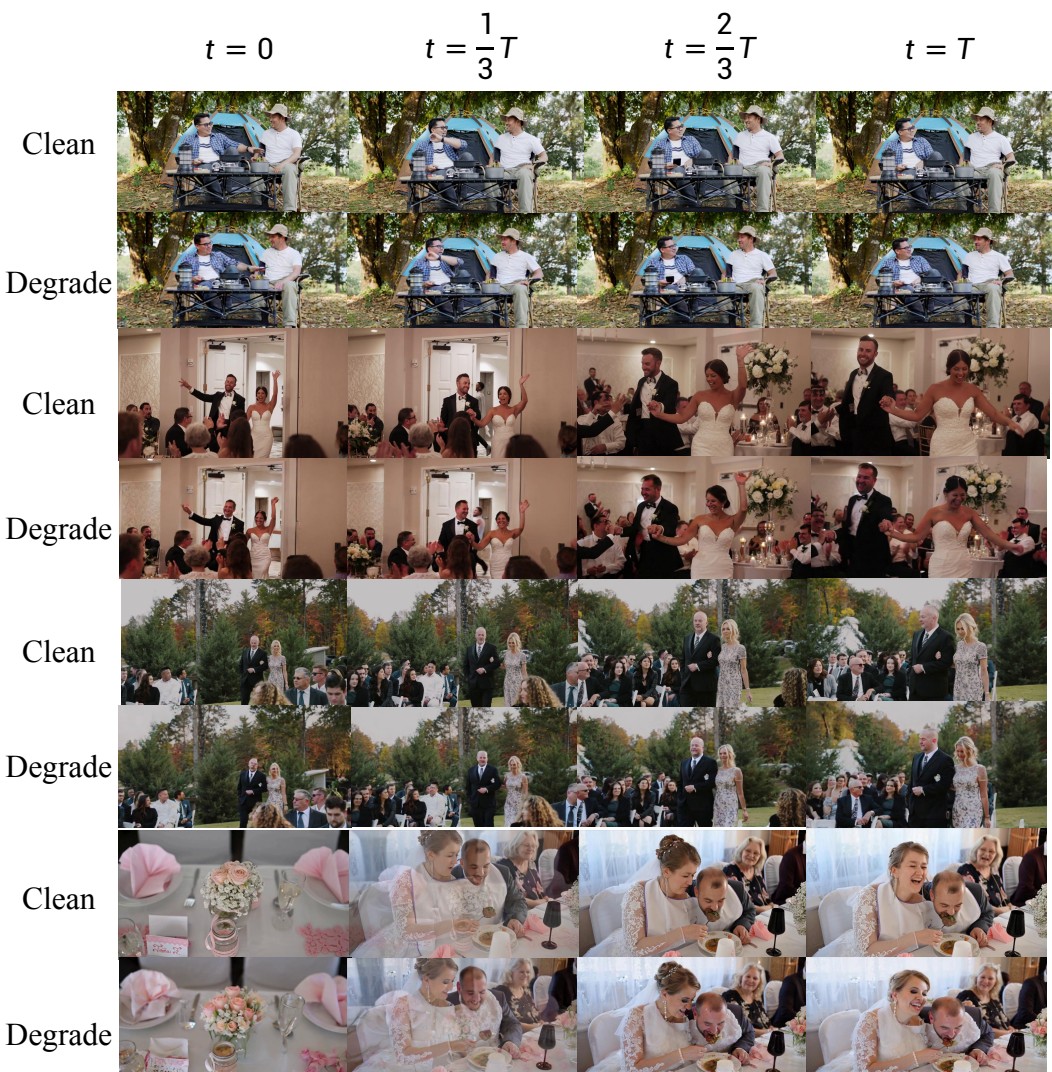

Figure 11: Samples of heavy sdedit degradation.

### A.5.2    QUANTITATIVE COMPARISONS

Full quantitative comparisons of text-to-video generation, multi-ID image-guided text-to-video generation and text-guided video editing tasks are shown in Tab. 5, 6 and 7 respectively.

### A.5.3    QUALITATIVE COMPARISONS

Additional qualitative comparisons of text-to-video generation, multi-ID image-guided text-to-video generation and text-guided video editing tasks are presented in Fig. 13, 14 and 15 respectively.

### A.5.4    ABLATION STUDY

Qualitative comparisons with different components are shown in Fig. 16. For architecture design, full channel-concat (Full CC) struggles to inject visual references. For full token-concat (Full TC), although it achieves comparable results, it largely sacrifices the inference efficiency. For degradation effect, sdedit-only and synthetic-only methods lack in generating vivid details and preserving input ID images respectively. For training order, both full training and easy-to-difficult paradigm show suboptimal results compared with our difficult-to-easy paradigm, which demonstrates the effectiveness of our training strategy.

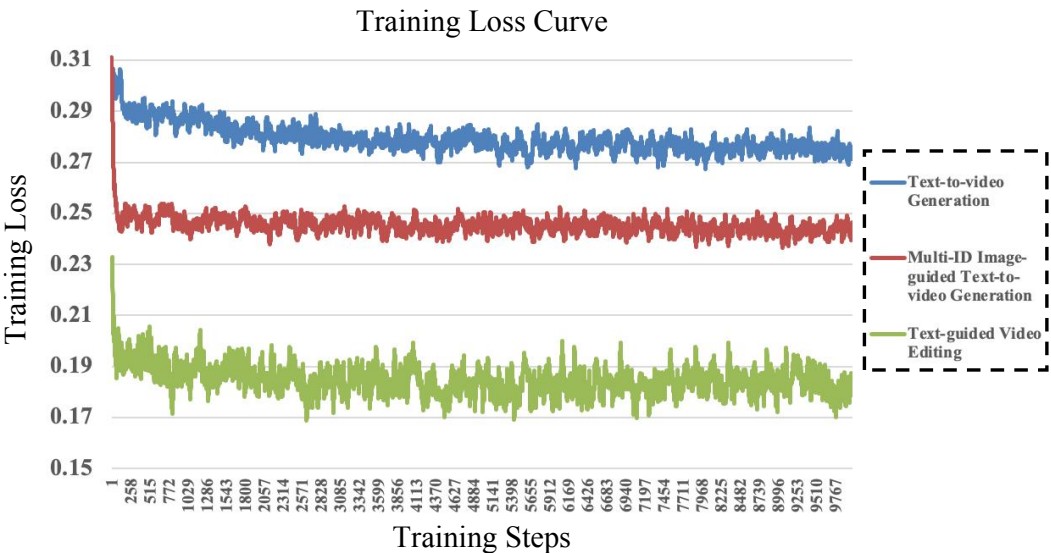

Figure 12: Training loss curve of all three tasks.

Table 5: Quantitative comparison of text-to-video generation task. **Bold** and underlined indicate the best and second-best results, respectively. ↑ indicates higher is better; ↓ indicates lower is better.

| Method | Visual Quality | | | | Subject Consistency | | Video Alignment | | |
|---|---|---|---|---|---|---|---|---|---|
| | **MUSIQ↑** | **CLIP-IQA↑** | **QAlign↑** | **DOVER↑** | **CLIP-I↑** | **DINO-I↑** | **PSNR↑** | **SSIM↑** | **LPIPS↓** |
| Base 512×512 | 30.996 | 0.246 | 3.741 | 0.594 | - | - | - | - | - |
| Base 1080P | 46.645 | 0.306 | 4.246 | 0.749 | - | - | - | - | - |
| VEnhancer-v1 | **57.171** | 0.367 | 4.214 | 0.733 | - | - | - | - | - |
| VEnhancer-v2 | 44.603 | 0.364 | 4.091 | 0.693 | - | - | - | - | - |
| STAR-light | 56.904 | 0.369 | 4.435 | 0.769 | - | - | - | - | - |
| STAR-heavy | 55.294 | 0.361 | **4.579** | **0.795** | - | - | - | - | - |
| SeedVR-7B | 55.596 | **0.379** | 4.396 | 0.778 | - | - | - | - | - |
| SeedVR-3B | 54.310 | 0.375 | 4.281 | 0.762 | - | - | - | - | - |
| SeedVR2-7B | 48.490 | 0.310 | 4.292 | 0.763 | - | - | - | - | - |
| SeedVR2-7B-sharp | 47.013 | 0.300 | 4.234 | 0.753 | - | - | - | - | - |
| SeedVR2-3B | 50.604 | 0.328 | 4.318 | 0.783 | - | - | - | - | - |
| Ours (single) | 56.146 | 0.366 | 4.535 | 0.771 | - | - | - | - | - |
| **Ours (unified)** | 56.418 | 0.371 | 4.500 | 0.778 | - | - | - | - | - |

The UniMMVSR model is initialized from a pre-trained text-to-video model, with the text embedding injected by a 2D cross-attention module per layer. To faster convergence, we maintain the same architecture design for text modality as the base model to avoid time-consumed re-training. We have conducted an additional quantitative comparison with token-concatenating text embedding design on text-to-video generation and multi-ID image-guided text-to-video generation tasks in Tab. 8. For both tasks, token-concatenating low-resolution video leads to comparable performance as our UniMMVSR architecture. However, as can be seen in the "CC LR video & TC others" and "Full TC" lines, directly token-concatenating text embedding results in severe performance degradation on both video quality and controlling metrics. We hypothesize that token-concatenating text embedding requires intensive re-training, which is prohibitive on a video super-resolution task.

### A.5.5 4K RESULTS

Additional 4K results of text-to-video generation, multi-ID image-guided text-to-video generation and text-guided video editing tasks are presented in Fig. 17, 18 and 19 respectively. The results show that the proposed cascaded framework excels at scaling resolution on all three controllable video generation tasks. We also present 4K videos in the supplementary material.

Table 6: Quantitative comparison of multi-ID image-guided text-to-video generation task. **Bold** and underlined indicate the best and second-best results, respectively. ↑ indicates higher is better; ↓ indicates lower is better.

| Method | Visual Quality | | | | Subject Consistency | | Video Alignment | | |
|---|---|---|---|---|---|---|---|---|---|
| | MUSIQ↑ | CLIP-IQA↑ | QAlign↑ | DOVER↑ | CLIP-I↑ | DINO-I↑ | PSNR↑ | SSIM↑ | LPIPS↓ |
| Base 512×512 | 29.314 | 0.255 | 3.149 | 0.433 | 0.692 | 0.538 | - | - | - |
| Base 1080P | 46.780 | 0.345 | 4.092 | 0.662 | 0.691 | 0.507 | - | - | - |
| VEnhancer-v1 | 60.656 | **0.469** | 4.149 | 0.707 | 0.671 | 0.533 | - | - | - |
| VEnhancer-v2 | 43.776 | 0.422 | 3.860 | 0.628 | 0.690 | 0.538 | - | - | - |
| STAR-light | 58.810 | 0.449 | 4.282 | **0.763** | 0.696 | 0.546 | - | - | - |
| STAR-heavy | 54.446 | 0.399 | 4.223 | 0.721 | 0.695 | 0.547 | - | - | - |
| SeedVR-7B | 54.491 | 0.419 | 3.960 | 0.708 | 0.693 | 0.543 | - | - | - |
| SeedVR-3B | 53.943 | 0.416 | 3.845 | 0.689 | 0.696 | 0.544 | - | - | - |
| SeedVR2-7B | 49.220 | 0.344 | 3.814 | 0.664 | 0.689 | 0.543 | - | - | - |
| SeedVR2-7B-sharp | 46.718 | 0.332 | 3.751 | 0.639 | 0.691 | 0.545 | - | - | - |
| SeedVR2-3B | 51.169 | 0.368 | 3.850 | 0.690 | 0.694 | 0.544 | - | - | - |
| Ours (no ref) | 60.947 | 0.445 | 4.385 | 0.742 | 0.693 | 0.543 | - | - | - |
| Ours (single) | 61.357 | 0.446 | 4.414 | 0.743 | **0.728** | **0.566** | - | - | - |
| **Ours (unified)** | **62.248** | 0.465 | **4.428** | 0.745 | 0.726 | **0.566** | - | - | - |

Table 7: Quantitative comparison of text-guided video editing task. **Bold** and underlined indicate the best and second-best results, respectively. ↑ indicates higher is better; ↓ indicates lower is better.

| Method | Visual Quality | | | | Subject Consistency | | Video Alignment | | |
|---|---|---|---|---|---|---|---|---|---|
| | MUSIQ↑ | CLIP-IQA↑ | QAlign↑ | DOVER↑ | CLIP-I↑ | DINO-I↑ | PSNR↑ | SSIM↑ | LPIPS↓ |
| Base 512×512 | 35.073 | 0.234 | 3.615 | 0.400 | - | - | 30.191 | 0.699 | 0.364 |
| Base 1080P | 53.616 | 0.383 | 4.247 | 0.634 | - | - | 29.383 | 0.582 | 0.358 |
| Ref Video | 54.249 | 0.365 | 4.131 | 0.571 | - | - | - | - | - |
| VEnhancer-v1 | 57.036 | 0.380 | 4.013 | 0.590 | - | - | 28.417 | 0.571 | 0.489 |
| VEnhancer-v2 | 48.084 | 0.353 | 3.959 | 0.557 | - | - | 28.712 | 0.628 | 0.410 |
| STAR-light | 56.802 | 0.397 | 4.264 | 0.608 | - | - | 29.421 | 0.631 | 0.397 |
| STAR-heavy | 56.207 | 0.378 | 4.259 | 0.599 | - | - | 29.442 | 0.648 | 0.371 |
| SeedVR-7B | 57.820 | 0.370 | 4.183 | 0.635 | - | - | 29.535 | 0.597 | 0.413 |
| SeedVR-3B | 55.326 | 0.360 | 4.048 | 0.628 | - | - | 29.338 | 0.588 | 0.416 |
| SeedVR2-7B | 54.046 | 0.361 | 4.087 | 0.610 | - | - | 29.600 | 0.614 | 0.367 |
| SeedVR2-7B-sharp | 52.723 | 0.359 | 4.010 | 0.579 | - | - | 29.563 | 0.619 | 0.362 |
| SeedVR2-3B | 54.310 | 0.355 | 4.099 | 0.597 | - | - | 29.326 | 0.593 | 0.382 |
| Ours (no ref) | **59.119** | **0.399** | 4.289 | **0.648** | - | - | 29.615 | 0.581 | 0.429 |
| Ours (single) | 53.388 | 0.348 | 4.302 | 0.597 | - | - | **31.905** | **0.723** | **0.276** |
| **Ours (unified)** | 53.245 | 0.344 | **4.305** | 0.597 | - | - | 31.556 | 0.713 | 0.282 |

### A.5.6 VIDEO RESULTS

Readers are recommended to check our provided video results in the supplementary material. Due to space limit, we have packed all previous video results in a single demo. We have additionally provided more 4K text-to-video generation and noise augmentation effect results to demonstrate the scalability and fidelity of our cascaded framework respectively.

### A.5.7 SELECTION OF THE RoPE INTERVAL

We follow two rules to assign the RoPE intervals: first, ref tokens of different categories do not overlap to avoid confusion; second, we assign smaller intervals for ref tokens of more difficult tasks to accelerate convergence. Specifically, we assign 0 to 20 for noise tokens, 20 to 30 for ref images and 30 to 50 for ref videos. To verify this, we have compare with an ablation (20 to 40 for ref images and 40 to 50 for ref videos) in Tab. 9.

As can be seen, switching the rope intervals results in comparable video quality. For the ablation model, even when a relatively close interval is assigned, its control capability on simple tasks (video editing) does not improve significantly; however, its control capability on complex tasks (multi-image) decreases to a certain extent as a relatively distant interval is assigned (0.720 to 0.726 for CLIP-I and 0.561 to 0.566 for DINO-I).

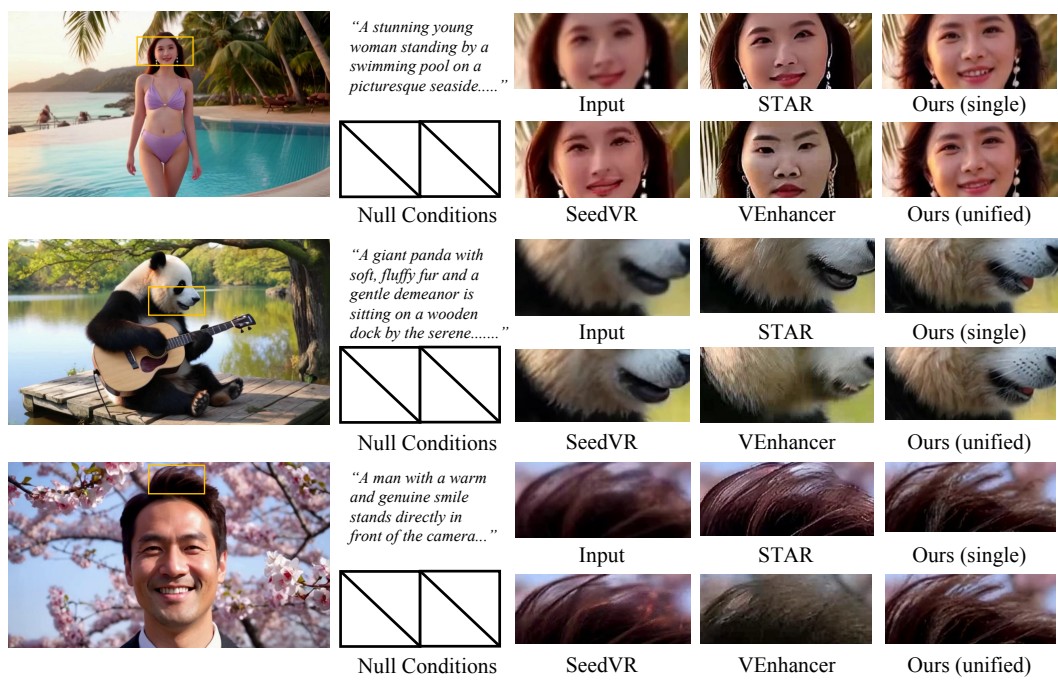

Figure 13: Qualitative comparisons on text-to-video generation task.

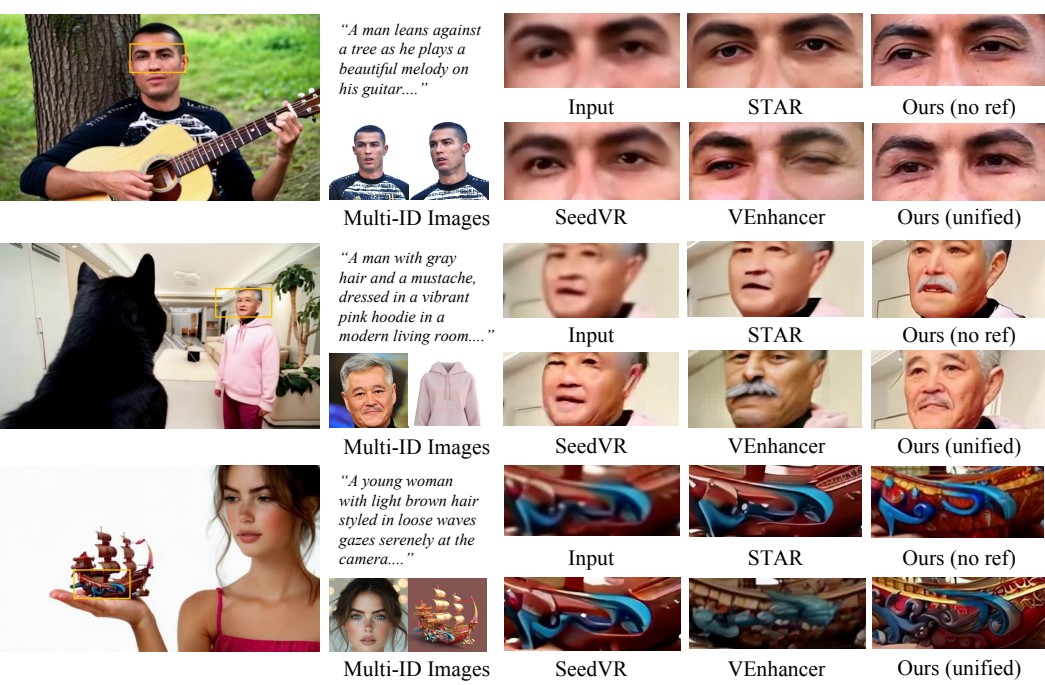

Figure 14: Qualitative comparisons on multi-ID image-guided text-to-video generation task.

### A.5.8 NOISE AUGMENTATION EFFECT

The motivation of UniMMVSR is to leverage multi-modal conditions (reference images/videos) to restore low-resolution videos in a guided manner. Therefore, we aim to remove the degraded pattern in the low-resolution videos and generate high-frequency details that are more "similar" to the referenced images/videos. For controllable generation task like multi-ID image-guided text-to-video

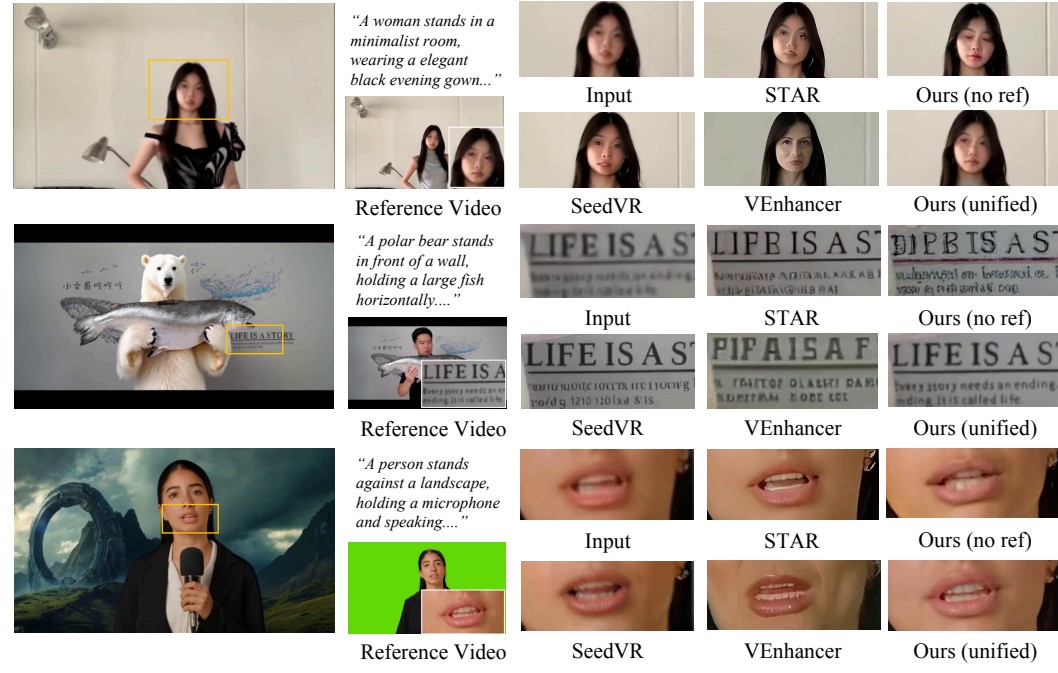

Figure 15: Qualitative comparisons on text-guided video editing task.

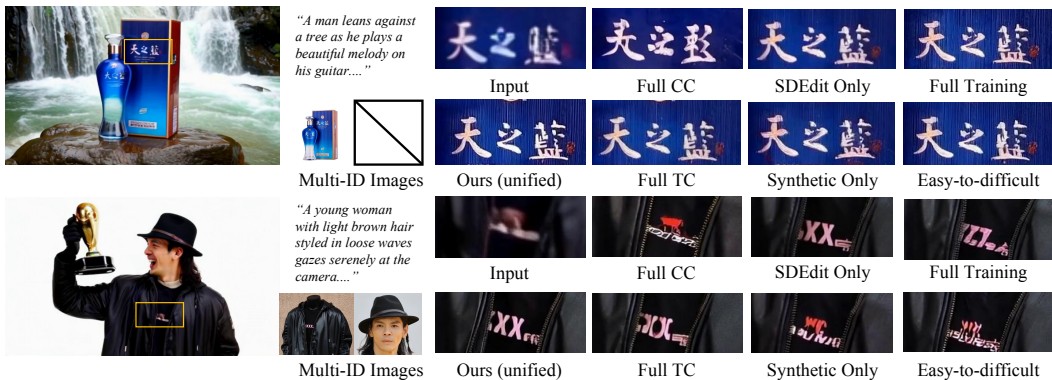

Figure 16: Qualitative comparisons with different components.

generation and text-guided video editing tasks, we generally do not consider the fidelity of the input video, as our primary goal is to restore it to achieve a higher degree of similarity with the reference condition. Thus, we set the noise augmentation timestep to 600 for stronger restoration ability (randomly select from 200 to 600 during training). The qualitative comparisons are shown in Fig. 20. As can be seen, by setting noise augmentation timestep to 200, our model successfully maintains global and local structures of the input video, demonstrating superior fidelity of our method for traditional video super-resolution task.

A.5.9 ROBUSTNESS AGAINST SUB-TASK PROBABILITY

The optimal sub-task probability might differ for different training data or model architecture. We set up the task probability following a principle that assigning more training steps for more difficult tasks. We have compared with different sub-task probability in Tab. 10.

For text-to-video task, further increasing the probability does not lead to a significant improvement. For multi-ID image-guided task, our architecture is robust to the task probability in a certain range (0.6 to 0.8). The result of t2v0.5&mi2v0.5 maintains a comparable quality to the settings used in our

Table 8: Additional ablation study of UniMMVSR with token-concatenating text embedding design. ↑ indicates higher is better; ↓ indicates lower is better. "CC" denotes channel-concat, "TC" denotes token-concat and "Cross" denotes 2D cross-attention.

| Method | Visual Quality | | | | Subject Consistency | |
|---|---|---|---|---|---|---|
| | MUSIQ↑ | CLIP-IQA↑ | QAlign↑ | DOVER↑ | CLIP-I↑ | DINO-I↑ |
| **Text-to-video Generation** | | | | | | |
| **Ours** | 56.422 | 0.375 | 4.510 | 0.780 | - | - |
| **Cross text&TC others** | 56.120 | 0.379 | 4.422 | 0.782 | - | - |
| **CC LR video&TC others** | 52.142 | 0.335 | 3.829 | 0.672 | - | - |
| **Full TC** | 49.235 | 0.307 | 3.555 | 0.669 | - | - |
| **Multi-ID Image-guided Text-to-video Generation** | | | | | | |
| **Ours** | 62.455 | 0.471 | 4.511 | 0.752 | 0.728 | 0.565 |
| **Cross text&TC others** | 62.399 | 0.488 | 4.582 | 0.751 | 0.730 | 0.562 |
| **CC LR video&TC others** | 54.314 | 0.369 | 3.434 | 0.647 | 0.675 | 0.525 |
| **Full TC** | 53.017 | 0.351 | 3.439 | 0.644 | 0.666 | 0.517 |

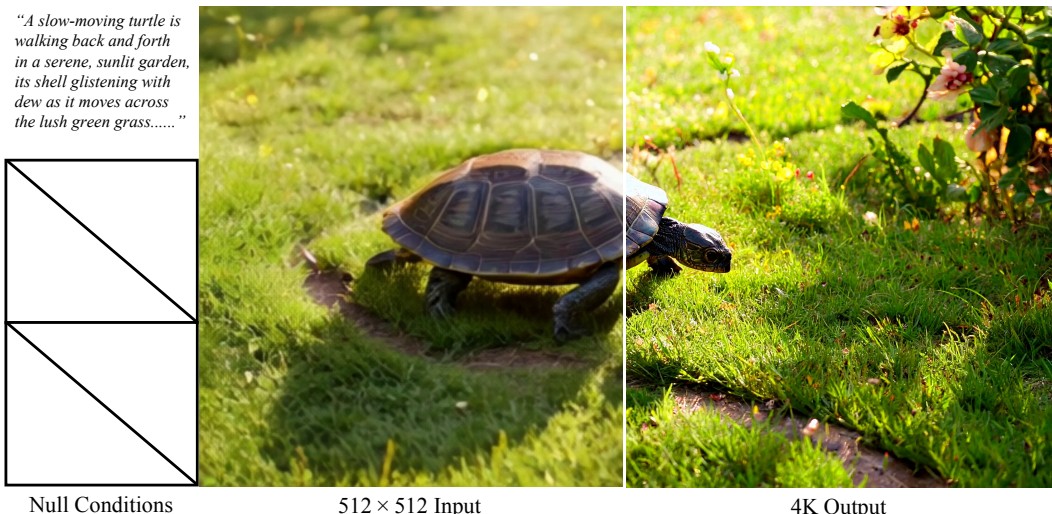

*"A slow-moving turtle is walking back and forth in a serene, sunlit garden, its shell glistening with dew as it moves across the lush green grass......"*

Null Conditions          512 × 512 Input          4K Output

Figure 17: Additional 4K results on text-to-video generation task. **Zoom-in for best view.**

paper. However, when the probability of t2v task is further reduced to 0.4, the model's performance on the t2v task begins to decline. We believe that for the mi2v task, the model can use the input reference conditions to generate new details in a guided manner, which reduces the difficulty of this task compared with the t2v task. The model only needs a certain amount of training data in the early stage to adapt to the reference image modality.

Although assigning prob=0.1 shows significant performance degradation, we hypothesize that the main reason is that the model has not converged, and increasing the number of training steps can effectively alleviate this issue. To confirm this, we have additionally trained 10k steps for this setting. The results are shown in Tab. 11. As can be seen, additionally training 10k steps leads to a significant improvement in the multi-ID image-guided text-to-video generation task, further validating the hypothesis of model converge. In general, if the number of training steps is sufficient, the training probability of different sub-tasks will not excessively affect the final quality of the model. Therefore, our exploration of the training probability is only to enable the model to converge faster across multiple sub-tasks.

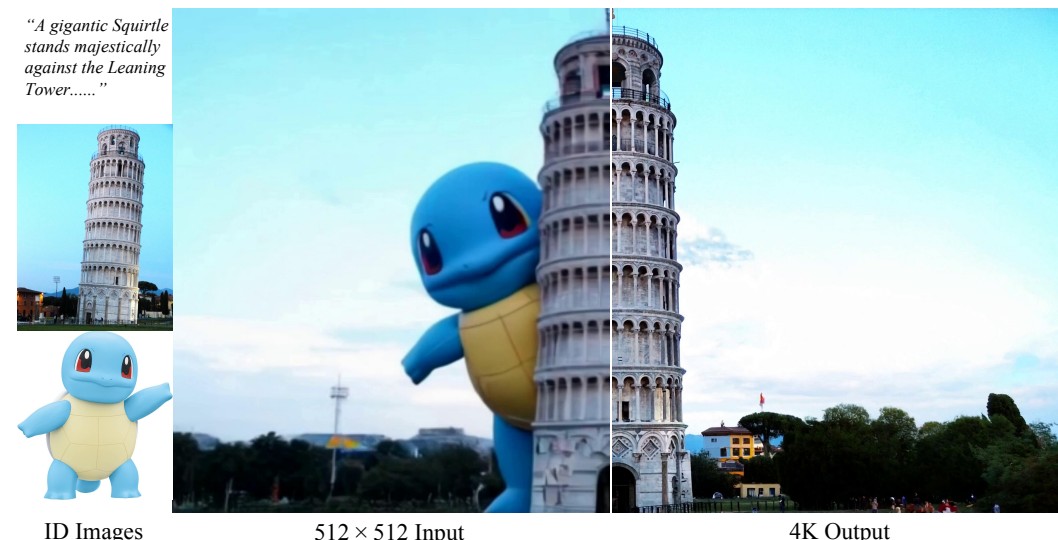

*"A gigantic Squirtle stands majestically against the Leaning Tower......"*

ID Images        512 × 512 Input        4K Output

Figure 18: Additional 4K results on multi-ID image-guided text-to-video generation task. **Zoom-in for best view.**

Table 9: Quantitative Evaluation of UniMMVSR with the ablation model about RoPE interval of the reference tokens. ↑ indicates higher is better; ↓ indicates lower is better.

| Method | Visual Quality | | | | Subject Consistency | | Video Alignment | | |
|---|---|---|---|---|---|---|---|---|---|
| | MUSIQ↑ | CLIP-IQA↑ | QAlign↑ | DOVER↑ | CLIP-I↑ | DINO-I↑ | PSNR↑ | SSIM↑ | LPIPS↓ |
| **Text-guided Video Editing** | | | | | | | | | |
| **Ours** | 53.245 | 0.344 | 4.305 | 0.597 | - | - | 31.556 | 0.713 | 0.282 |
| **Ablation** | 53.242 | 0.345 | 4.310 | 0.595 | - | - | 31.555 | 0.715 | 0.281 |
| **Multi-ID Image-guided Text-to-video Generation** | | | | | | | | | |
| **Ours** | 62.248 | 0.465 | 4.428 | 0.745 | 0.726 | 0.566 | - | - | - |
| **Ablation** | 62.210 | 0.462 | 4.412 | 0.744 | 0.720 | 0.561 | - | - | - |

### A.5.10 COMPUTATIONAL COMPLEXITY ANALYSIS

We conduct a detailed comparison of computational complexity on the multi-ID image-guided task. The comparison is performed on a single H800 GPU for a 1080P video with 77 frames. We keep other settings the same as the official repo for other baselines to maintain high-quality generation results of these methods. The results are shown in Tab. 12. From these results, it can be seen that our 10B version achieves the optimal video quality and controlling ability with only a negligible increase in computational cost. Meanwhile, for our 1B version, although significantly reducing the model size leads to a decrease in controlling metrics, it still outperforms other baselines, demonstrating the effectiveness of the proposed framework.

### A.5.11 COMPARISONS WITH A SMALL BACKBONE

For the results shown in the main text, we use a pre-trained 10B text-to-video model as initialization to obtain better reference condition injection capability. Also, we claim that the proposed framework can be directly applied to a smaller backbone without modification. To confirm this, we replicate our framework using a pre-trained 1B t2v model as initialization. The quantitative comparison can be seen in Tab. 12. For our 1B version, although significantly reducing the model size leads to a decrease in controlling metrics, it still outperforms other baselines, demonstrating the effectiveness of the proposed framework.

*"A person stands in a vast, open field with rolling hills in the background. they are dressed in traditional attire, holding a large, majestic dragon on their gloved hand......"*

| Reference Video | 512 × 512 Input | 4K Output |

Figure 19: Additional 4K results on text-guided video editing task. **Zoom-in for best view.**

### A.5.12 USER STUDY

For further comprehensive comparisons, we carried out a user study that evaluated the results of both text-to-video generation and multi-ID image-guided text-to-video generation tasks. We included all three baselines in this study, consisting of SeedVR, STAR and VEnhancer. We invite a total of 20 participants for this user study. Each volunteer was presented with a set of 10 randomly selected video triplets, which included an input video, the results obtained from the compared methods, and our result. For the 10 video triplets, each of the two aforementioned tasks accounts for half. For the text-to-video generation task, their role was to choose the visually superior enhanced video from the given options. For the multi-ID image-guided text-to-video generation task, their role was to choose the visually similar video compared with the referenced images. The results are shown in Tab. 13. From the results, it can be seen that our method generates more natural details for AIGC input than other baselines. For multi-ID image-guided text-to-video generation task, results reveal a clear preference of controll ability among the volunteers for our results over those produced by other methods.

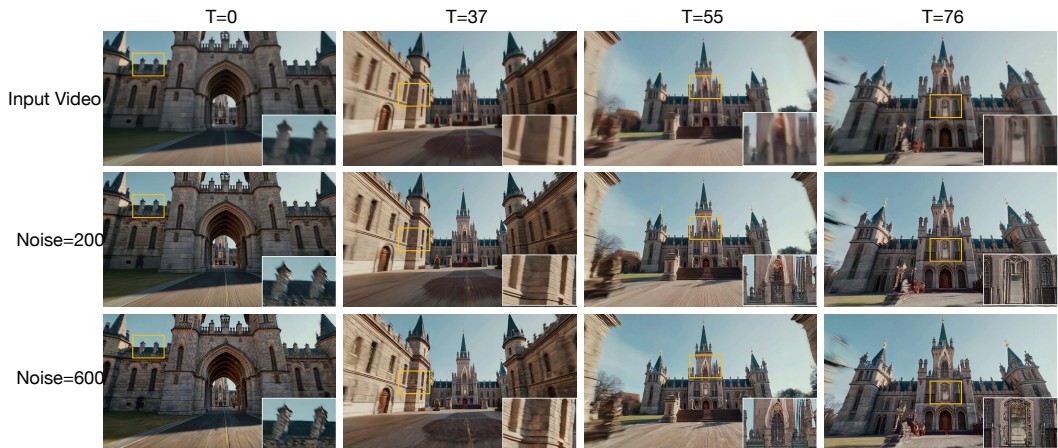

Figure 20: Qualitative comparisons of noise augmentation effect.

Table 10: Quantitative evaluation of UniMMVSR with different sub-task probability. ↑ indicates higher is better; ↓ indicates lower is better. "t2v" and "mi2v" refer to text-to-video generation and multi-ID image-guided tasks respectively.

| Method | Visual Quality | | | | Subject Consistency | |
|---|---|---|---|---|---|---|
| | MUSIQ↑ | CLIP-IQA↑ | QAlign↑ | DOVER↑ | CLIP-I↑ | DINO-I↑ |
| **Text-to-video Generation** | | | | | | |
| **t2v0.4&mi2v0.6** | 55.278 | 0.359 | 4.302 | 0.755 | - | - |
| **t2v0.5&mi2v0.5** | 56.321 | 0.376 | 4.500 | 0.779 | - | - |
| **t2v0.6&mi2v0.4 (ours)** | 56.422 | 0.375 | 4.510 | 0.780 | - | - |
| **t2v0.7&mi2v0.3** | 56.245 | 0.377 | 4.522 | 0.778 | - | - |
| **t2v0.8&mi2v0.2** | 56.344 | 0.374 | 4.499 | 0.781 | - | - |
| **t2v0.9&mi2v0.1** | 56.445 | 0.378 | 4.503 | 0.779 | - | - |
| **Multi-ID Image-guided Text-to-video Generation** | | | | | | |
| **t2v0.4&mi2v0.6** | 62.527 | 0.479 | 4.504 | 0.754 | 0.726 | 0.567 |
| **t2v0.5&mi2v0.5** | 62.578 | 0.474 | 4.492 | 0.750 | 0.727 | 0.567 |
| **t2v0.6&mi2v0.4 (ours)** | 62.455 | 0.471 | 4.511 | 0.752 | 0.728 | 0.565 |
| **t2v0.7&mi2v0.3** | 62.271 | 0.465 | 4.488 | 0.755 | 0.727 | 0.563 |
| **t2v0.8&mi2v0.2** | 62.344 | 0.469 | 4.434 | 0.749 | 0.725 | 0.560 |
| **t2v0.9&mi2v0.1** | 58.007 | 0.421 | 3.839 | 0.688 | 0.691 | 0.529 |

## A.6 DETAILS OF TEST SET

Based on the base model output, we collect a dataset, MMVSR90, which comprises 30 video clips per task. To ensure the diversity of the test set, this dataset covers a wide range of scenarios, including different subjects (e.g., humans and animals), various camera motions, and diverse backgrounds. To provide more details, we have presented the category composition and motion-intensity distribution of the MMVSR90 dataset in Fig. 21. It encompasses a wide variety of semantic categories and spans a broad spectrum of motion intensities, providing a comprehensive benchmark for evaluating model performance.

## A.7 LIMITATIONS

To demonstrate the resolution scaling ability of the cascaded framework, we have collected 3.3w 4K videos with around 1 second per task. Due to data limit, some 4K generation results might occur slight degradation, which origins from model under-fitting. Further expanding the capacity of the 4K dataset and extending its duration to 5 seconds will be the direction for the next phase.

Table 11: Additional quantitative evaluation of UniMMVSR with t2v0.9&mi2v0.1 setting across different training steps. ↑ indicates higher is better; ↓ indicates lower is better. "t2v" and "mi2v" refer to text-to-video generation and multi-ID image-guided tasks respectively.

| Method | Visual Quality | | | | Subject Consistency | |
|---|---|---|---|---|---|---|
| | MUSIQ↑ | CLIP-IQA↑ | QAlign↑ | DOVER↑ | CLIP-I↑ | DINO-I↑ |
| **Text-to-video Generation** | | | | | | |
| **t2v0.6&mi2v0.4 (ours)** | 56.422 | 0.375 | 4.510 | 0.780 | - | - |
| **t2v0.9&mi2v0.1 (10k steps)** | 56.445 | 0.378 | 4.503 | 0.779 | - | - |
| **t2v0.9&mi2v0.1 (20k steps)** | 56.952 | 0.385 | 4.498 | 0.782 | - | - |
| **Multi-ID Image-guided Text-to-video Generation** | | | | | | |
| **t2v0.6&mi2v0.4 (ours)** | 62.455 | 0.471 | 4.511 | 0.752 | 0.728 | 0.565 |
| **t2v0.9&mi2v0.1 (10k steps)** | 58.007 | 0.421 | 3.839 | 0.688 | 0.691 | 0.529 |
| **t2v0.9&mi2v0.1 (20k steps)** | 59.952 | 0.444 | 4.127 | 0.710 | 0.709 | 0.544 |

Table 12: Computational complexity analysis of UniMMVSR with other baselines on the multi-ID image-guided text-to-video generation task. ↑ indicates higher is better; ↓ indicates lower is better.

| Method | Visual Quality | | | | Subject Consistency | | Computational Efficiency | | |
|---|---|---|---|---|---|---|---|---|---|
| | MUSIQ↑ | CLIP-IQA↑ | QAlign↑ | DOVER↑ | CLIP-I↑ | DINO-I↑ | Params (B) | Time (s) | Memory (GB) |
| **Ours (10B)** | 62.248 | 0.465 | 4.428 | 0.745 | 0.726 | 0.566 | 10.0 | 1030.552 | 76.530 |
| **Ours (1B)** | 61.278 | 0.458 | 4.382 | 0.733 | 0.719 | 0.562 | 1.0 | 237.931 | 46.258 |
| **SeedVR** | 54.491 | 0.419 | 3.960 | 0.708 | 0.693 | 0.543 | 7.0 | 630.286 | 70.568 |
| **STAR** | 58.810 | 0.449 | 4.282 | 0.763 | 0.696 | 0.546 | 2.0 | 555.347 | 62.220 |
| **VEnhancer** | 60.656 | 0.469 | 4.149 | 0.707 | 0.671 | 0.533 | 2.0 | 542.356 | 60.554 |

Moreover, the generalization of the current framework to datasets of different sub-tasks has not yet been fully verified. Due to the difficulty of data collection, the ablation experiments on the generalization of the dataset are still in preparation at present. Exploring the generalization of the proposed framework to diverse datasets will be another direction in the next phase.

## A.8 ETHICS STATEMENT

This work includes a user study involving human subjects. All participants were informed of the study's purpose and provided consent prior to participation. The study design and procedures were conducted in a manner consistent with ethical standards to ensure the protection of participants' rights and privacy. In addition, as with any generative model, our method carries the risk of potential misuse. We emphasize that the system should be applied responsibly and urge caution to avoid malicious or harmful applications.

## A.9 REPRODUCIBILITY STATEMENT

To ensure the reproducibility of our work, we will ensure the following points. Code: Our code and model will be made publicly available, including necessary scripts. Data: Detailed descriptions of our data processing are provided in Supp. A.2. Experimental Setup: We have stated all experimental configurations, including hyperparameters, hardware specifications in Sec. 4.1 of the main text. Model Architecture: The architecture details are described in method part.

Table 13: User Study of UniMMVSR with other three baselines on text-to-video generation and multi-ID image-guided text-to-video generations tasks.

| Task | Ours | SeedVR | STAR | VEnhancer |
|---|---|---|---|---|
| Text-to-video Generation | 0.68 | 0.08 | 0.16 | 0.08 |
| Multi-ID Image-guided Text-to-video Generation | 0.91 | 0.06 | 0.02 | 0.01 |

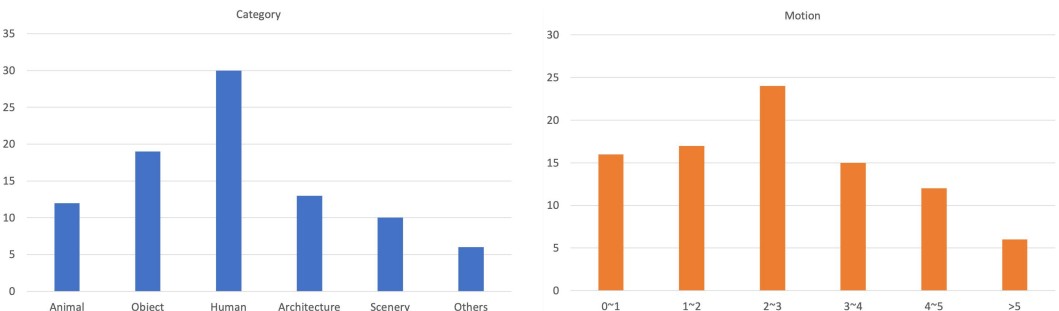

Figure 21: Statistical analysis of categories and motion in MMVSR90.

