# OpenReview forum: "UniMMVSR: A Unified Multi-Modal Framework for Cascaded Video Super-Resolution"
_ICLR.cc/2026/Conference — Submitted to ICLR 2026_

### Official Review · Reviewer_YT7R · 2025-10-17

**Soundness:** 3
**Presentation:** 2
**Contribution:** 2
**Rating:** 4
**Confidence:** 5

**Summary:**

This paper introduces UniMMVSR, a unified multi-modal generative video super-resolution framework designed to upscale AI-generated low-resolution videos to ultra-high-resolution (e.g., 4K), supporting multi-model input as guidance, including text, multi-ID images, and reference videos.
The proposed solution includes the following parts:
- A unified conditioning strategy combining channel concatenation (for low-res video) and token concatenation (for reference modalities), with separated conditional RoPE for positional embedding.
- A SDEdit-based degradation pipeline that simulates realistic generative degradations of base models, combining with the conventional synthetic degradation baseline.
- A difficult-to-easy curriculum training strategy to align sub-tasks of varying complexity.

**Strengths:**

- This paper introduces multi-modal guidance into diffusion-based video super-resolution (VSR). The qualitative results demonstrate the effectiveness of the guidance in guiding the content generation with appealing performance.
- The ablation study shows the effectiveness of the proposed sdedit degradation as well as the the conditioning strategy for multi-modal guidance VSR

**Weaknesses:**

- The contributions of the paper seem to weigh more on the engineering side rather than the technical novelty. This paper seems to be built on lots of existing successful practices, like DiT, RoPE, cascaded modeling, etc. The improvements upon the above parts, e.g., conditioning strategy, degradation pipeline and training order seem kind of trival, though maybe effective.
- The fidelity of the model seems poor compared with previous baselines like SeedVR and STAR, which is obvious in nearly all the qualitative results. It is unclear if this is caused by the introduced multi-model guidance.

**Questions:**

My main concerns of this paper are as follows:

1. The two major weaknesses above, i.e., technical contributions and fidelity.

2. Some of the paper details are vague: 1) Achieving 4K SR is very VRAM expensive. The details of how to achieve such high-resolution video need further explanation, e.g., how many frames can be processed in one inference? how to process long videos? How much VRAM is required for each inference, etc? 2) The details of image-related transformations for reference alignment are vague in Line 299-300. What detailed transformations are used for training?

3. The complexity comparison is missing. There should be FLOPs, parameters and inference time comparison. The training time should also be provided.

4. The details of the evaluation data seem to be missing. What data is used for quantitative and qualitative comparison? The author may also consider making comparisons on commonly used synthetic and real-world benchmarks, following previous works such as Upsale-A-Video, STAR, and SeedVR, with standard metrics such as PSNR, SSIM, CLIP-IQA, MUSIQ, etc. This ensures a more straightforward comparison with previous methods.

**Details Of Ethics Concerns:**

NA.

---

> ### Author Response · Authors · 2025-11-26
> **Official Comment by Authors (Part 1/3)**
>
> We sincerely appreciate this reviewer's insightful feedback and careful review of our manuscript. Please check our detailed point-to-point responses to all the comments of this reviewer. We hope our responses can address this reviewer's concerns.
>
> ---
>
> `[W1] The architecture novelty of UniMMVSR.`
>
> Thanks for your valuable comment. We agree that channel-concat and token concat have been proposed in previous methods. However, we would like to clarify that we have not regarded these two technologies as the main novelty of this paper. Instead, the main contributions of this paper are as follows: We have identified that the fundamental reason why existing cascaded generation frameworks are hardly applicable to various controllable generation tasks lies in the randomness introduced during the sampling process of traditional super-resolution models. Therefore, we have tackled this challenge by introducing multi-modal reference conditions to the video super-resolution model, and introduced a new task--multimodal video super-resolution.
>
> Moreover, we have found that the reference distortion problems are usually observed in the base model output, which typically does not exist in traditional video super-resolution tasks. Therefore, we follow the principle ``the reference distortion is equivalent to the results obtained by base model using only text condition'' and introduce a new SDEdit degradation pipeline to effectively simulate this scenario.
>
> By constructing a unified multi-task framework, we have made an influential discovery in the academic community: high-quality data can be transferred across different sub-tasks. We believe this discovery will help reduce the difficulty of data collection for complex reference tasks.
>
> ---
> `[W2] The fidelity of UniMMVSR.`
>
> Thanks for your comment. The motivation of UniMMVSR is to leverage multi-modal conditions (reference images/videos) to restore low-resolution videos in a guided manner. Therefore, we aim to remove the degraded pattern in the low-resolution videos and generate high-frequency details that are more ``similar'' to the referenced images/videos. For controllable generation task like multi-ID image-guided text-to-video generation and text-guided video editing tasks, we generally do not consider the fidelity of the input video, as our primary goal is to restore it to achieve a higher degree of similarity with the reference condition. Thus, we set the noise augmentation timestep to 600 for stronger restoration ability (randomly select from 200 to 600 during training).
>
> To demonstrate the fidelity of our framework for traditional video super-resolution task, we perform a qualitative comparison of noise augmentation effect in Supp.A.5.8. As can be seen, by setting noise augmentation timestep to 200, our model successfully maintains global and local structures of the input video, demonstrating superior fidelity of our method for traditional video super-resolution task. We also provide the video results in the supplementary material. Please check them for details.

---

> ### Author Response · Authors · 2025-11-26
> **Official Comment by Authors (Part 2/3)**
>
> `[W3] Some of the paper details are vague.`
>
> Thanks for your valuable comment. For 4K video generation, we perform experiments to demonstrate the resolution scaling ability of our cascaded framework. Therefore, we train our model on 3.3w 4K videos with 21 frames per task. The memory usage is approximate as inferring a 1080P video with 77 frames (the main experiment setting). For this paper, we only consider designing a multi-modal video super-resolution model upon a multi-modal base model. Extending to long videos will be the direction for the next phase. The inference VRAM is 76.530GB, which can be afforded by modern A800/H800 GPUs.
>
> For image-related transform, we have added the details in the Multi-ID Image-guided Text-to-video Generation part of Supp.A.2, which comprises a random selection of brightness, flip, shearing, rotate and random crop during training. Details are shown as below (h and w denote the height and width of the reference image). The training code will be made public.
>
> |         Transform          |   Probability  |   Type  | Sampling Range |
> |:------------------------:|:---------:|:---------:|:---------:|
> |       Brightness       |   1.0   |   scale   | [0.9, 1.1] |
> |       Horizontal flip       |   0.5   |   -   | - |
> |       Shearing (x-axis)       |   1.0   |   value (px)   | [-0.05, 0.05]*w |
> |       Shearing (y-axis)       |   1.0   |   value (px)   | [-0.05, 0.05]*h |
> |       Rotation       |   1.0   |   value (degree)    | [-20, 20] |
> |       Random crop       |   0.5   |   scale   | [0.67, 1.0] |
>
> ---
> `[W4] Computational complexity analysis.`
>
> Thank you for your valuable comment. We agree that the most consuming part would lie in the training phase. Meanwhile, we also indicate that this is precisely one of the side advantages of the method we proposed: UniMMVSR leverages visual reference conditions to generate corresponding high-frequency details in a guided manner, which reduces the difficulty of model fitting and prevents the model from generating random textures using text. The training phase contains 3 stages, and it takes around 1 day per stage.
>
> For the results shown in the main text, we use a pre-trained 10B text-to-video model as initialization to obtain better reference condition injection capability. Also, we claim that the proposed framework can be directly applied to a smaller backbone without modification. To confirm this, we replicate our framework using a pre-trained 1B t2v model as initialization and conduct a detailed comparison on the multi-ID image-guided task. The comparison is performed on a single H800 GPU for a 1080P video with 77 frames.  We keep other settings the same as the official repo for other baselines to maintain high-quality generation results of these methods. The results are shown as follows:
>
> |          Method         | MUSIQ | CLIP-IQA | QAlign | DOVER |   CLIP-I   |   DINO-I  | Params (B) | Time (s) | Memory (GB) |
> |:------------------------:|:---------:|:---------:|:---------:|:---------:|:---------:|:---------:|:---------:|:---------:|:---------:|
> |       Ours (10B)       |   62.248   |   0.465   | 4.428 | 0.745 | 0.726 | 0.566 | 10.0 | 1030.552 | 76.530 |
> |       Ours (1B)       |   61.278   |   0.458   | 4.382 | 0.733 | 0.719 | 0.562 | 1.0 | 237.931 | 46.258 |
> |       SeedVR       |   54.491   |   0.419   | 3.960 | 0.708 | 0.693 | 0.543 | 7.0 | 630.286 | 70.568 |
> |       STAR       |   58.810   |   0.449   | 4.282 | 0.763 | 0.696 | 0.546 | 2.0 | 555.347 | 62.220 |
> |       VEnhancer      |   60.656   |   0.469   | 4.149 | 0.707 | 0.671 | 0.533 | 2.0 | 542.346 | 60.554 |
>
> From these results, it can be seen that our 10B version achieves the optimal video quality and controlling ability with only a negligible increase in computational cost. Meanwhile, for our 1B version, although significantly reducing the model size leads to a decrease in controlling metrics, it still outperforms other baselines, demonstrating the effectiveness of the proposed framework.

---

> ### Author Response · Authors · 2025-11-26
> **Official Comment by Authors (Part 3/3)**
>
> `[W5] Details of test set.`
>
> Thanks for your valuable comment. Based on the base model output, we collect a dataset, MMVSR90, which comprises 30 video clips per task. To ensure the diversity of the test set, this dataset covers a wide range of scenarios, including different subjects (e.g., humans and animals), various camera motions, and diverse backgrounds. To provide more details, we have presented the category composition and motion-intensity distribution of the MMVSR90 dataset in Fig.21. It encompasses a wide variety of semantic categories and spans a broad spectrum of motion intensities, providing a comprehensive benchmark for evaluating model performance. Moreover, we have already used standard metrics to evaluate the video quality (MUSIQ, CLIP-IQA, QAlign, DOVER) and controll ability (CLIP-I, DINO-I, PSNR, SSIM, LPIPS) in Tab.1.
>
> We are the first to introduce multi-modal video super-resolution paradigm to the cascaded framework. Thus, there is limited benchmark for this controllable scenario for comprehensive comparison. Since the MMVSR90 test set is collected from internal data, we are submitting it for review and will make it public in the next phase to further broadcast this task in the community.
>
> ---
> We hope the above statements can address your concerns. If you have any further questions, please let us know at any time.

---

> > ### Comment · Reviewer_YT7R · 2025-11-28
> >
> > I have carefully read the feedback from the authors and other reviewers. I appreciate the additional details and experiments provided by the authors. While the theoretical contribution of this work remains a concern from my view, the paper presents potential for practical applications. I hereby will raise my score to 6 later.

---

> > > ### Author Response · Authors · 2025-11-28
> > > **Official Comment by Authors**
> > >
> > > Dear Reviewer  YT7R,
> > >
> > > We are truly encouraged and grateful for your positive feedback and your decision to raise the rating to 6.
> > >
> > > We are delighted to learn that our detailed response and the new experimental results have successfully resolved your main concerns. Your constructive feedback throughout the review process has been invaluable in helping us identify areas for improvement and strengthen the validation of our method.
> > >
> > > Thank you once again for your time, support, and contribution to improving our work.
> > >
> > > Best regards,
> > >
> > > The Authors of Submission 1451

---

### Official Review · Reviewer_jTBQ · 2025-10-22

**Soundness:** 3
**Presentation:** 3
**Contribution:** 2
**Rating:** 4
**Confidence:** 4

**Summary:**

The paper proposes a cascaded video super-resolution framework that can support unified inputs of text, images, and videos. To unify different conditions together, the paper explores options such as channel concatenation of low-resolution video, token concatenation of visual references, and separated conditional RoPE for multi-ID images and reference videos. The paper also designs a degradation framework to simulate the artifacts and distortion generated by the base model. The contributions of this work are the following. The proposed framework is the first unified one that can handle multimodal conditions. The paper also provides some useful insights on how to effectively utilize diverse inputs—including low-resolution video, text, multiple ID images, and reference videos. The experimental demonstrations also look solid as it provides sufficient visualizations.

**Strengths:**

The strengths of this work are outlined below.

The proposed framework was claimed to be the first one that can handle multimodal conditions simultaneously for the video super-resolution task.

The paper provides some useful insights on how to effectively utilize diverse inputs—including low-resolution video, text, multiple ID images, and reference videos. Considering the difference in modalities, the paper explores options such as channel/token concatenation and separated RoPE.

The experimental demonstrations also look solid as they provide sufficient visualizations, where the ablation shows effectiveness for different components.

**Weaknesses:**

Though the paper offers the above strengths, it still demonstrates a few critical weaknesses, which need to be further clarified.

1. The entire framework looks a bit complicated. For different modalities, it chooses different options such as channel/token concatenation and separated RoPE.  Can we also process them all as tokens and then use token concatenation? After all, the paper shows that the token concatenation performs the best on multi-modal condition injection when incorporating multiple ID images and reference videos. In this case, the motivation work regarding this part could be further clarified.

2. The framework also relies on the choice of different training orders, basically on how to reorganize the order for different tasks of text-to-video, multi-ID image-guided, and video editing. This also depends on how to set up the probability for choosing different tasks. This makes the proposed framework not generalizable and could be difficult to train. It is unclear whether the distribution of these probability values will vary again if datasets and or some base architectures change.

3. The paper should provide some computational complexity analysis on the proposed framework, as it consists of many different components. The most consuming part would lie in the training phase. But it will still be good to list the inference time and the used parameters, and so on.

4. The reason for using multi-ID images is not well clarified. It is unknown which synthesis output relies on these multi-ID images. And among the provided multi-ID images, which image is placed with more emphasis?

5. The proposed method does not seem to achieve the best among the compared methods. For some listed methods such as VEnhancer, STAR and SeedVR, they can all handle the tasks of Text-to-video Generation, Text-guided Video Editing, and Multi-ID Image-guided Text-to-video Generation. What are the advantages of the proposed framework compared to these methods from this perspective?

**Questions:**

1. When describing the technical details of low-resolution video via channel concatenation, it appears that "Upsampler" was directly applied to the latent code, other than in the pixel space. There could be some confusion here.

2. The role of using multi-ID images needs to be better clarified. Based on the illustrations from Figure 3, why does the super-resolution on the text image rely on the input face image, e.g., the third row case? It appears that super-resolution output has nothing to do with the face image.

3. What is the computational complexity of the proposed method, and how reproducible is the proposed approach?

4. What is the motivation for not using the all tokens process for text, image, and videos? Has this been explored in this paper?

---

> ### Author Response · Authors · 2025-11-26
> **Official Comment by Authors (Part 1/4)**
>
> We sincerely appreciate this reviewer's insightful feedback and careful review of our manuscript. Please check our detailed point-to-point responses to all the comments of this reviewer. We hope our responses can address this reviewer's concerns.
>
> ---
> `[W1] The entire framework looks a bit complicated.`
>
> Thanks for your valuable comment. We have already performed ablation study in Tab.2 to explore the unified architecture. As can be seen, token-concatenating input video with other tokens exhibits comparable performance with our UniMMVSR architecture. However, since the computational cost of the 3D attention module increases quadratically with the number of tokens, we ultimately channel-concatenate the input video to accelerate the inference speed.
>
> To avoid the fidelity degradation caused by the randomness in the sampling process of super-resolution model within previous cascaded framework, we leverage the injection of multi-modal reference conditions to guide the generation of high-fidelity details and textures. The latent generated by the super-resolution model needs to extract multi-granular conditional information from the references. However, directly performing channel-concatenation between the references and the noise latent can only achieve coarse-grained matching and is prone to confusing the model, as shown in Tab.2. Therefore, we adopt a full attention mechanism: through a unified attention module, we use attention scores to automatically assign the importance of different reference conditions to the noise latent, thereby realizing more fine-grained injection.
>
> ---
> `[W2] The framework replies on the training order and probability.`
>
> Thank you for raising this important point. We agree that the optimal sub-task probability might differ for different training data or model architecture. We set up the task probability following a principle that assigning more training steps for more difficult tasks. Following your suggestion, we have compared with different sub-task probability as follows (t2v and mi2v refer to text-to-video generation and multi-ID image-guided tasks respectively):
>
> |          Ablation          |   MUSIQ   |   CLIPIQA  |    QAlign   |  DOVER | CLIPI | DINOI |
> |:------------------------:|:---------:|:---------:|:---------:|:---------:|:----------------:|:---------------:|
> |              |      |      |  Text-to-video Generation    |      |             |            |
> |       t2v0.6&mi2v0.4 (ours)       |   56.422   |   0.375   |   4.510   |   0.780   |       -      |      -      |
> |       t2v0.7&mi2v0.3      |   56.245   |   0.377   |   4.522   |   0.778   |       -      |      -      |
> |       t2v0.8&mi2v0.2      |   56.344   |   0.374   |   4.499   |   0.781   |       -      |      -      |
> |       t2v0.9&mi2v0.1      |   56.445   |   0.378   |   4.503   |   0.779   |       -      |      -      |
> |              |      |      |  Multi-ID Image-guided Text-to-video Generation    |      |             |            |
> |       t2v0.6&mi2v0.4 (ours)       |   62.455   |   0.471   |   4.511   |   0.752   |       0.728      |     0.565      |
> |       t2v0.7&mi2v0.3      |   62.271   |   0.465   |   4.488   |   0.755   |       0.727      |     0.563      |
> |       t2v0.8&mi2v0.2      |   62.344   |   0.469   |   4.434   |   0.749   |      0.725      |      0.560      |
> |       t2v0.9&mi2v0.1      |   58.007   |   0.421   |   3.839   |   0.688   |       0.691      |      0.529      |
>
> For text-to-video task, further increasing the probability does not lead to a significant improvement. For multi-ID image-guided task, our architecture is robust to the task probability in a certain range (0.6 to 0.8). Although assigning prob=0.1 shows significant performance degradation, we hypothesize that the main reason is that the model has not converged, and increasing the number of training steps can effectively alleviate this issue.

---

> > ### Comment · Reviewer_jTBQ · 2025-11-27
> >
> > Thanks for providing the experimental results on the training order and its associated probabilities. This does not seem to fully address the concerns, as it did not show the generalizability of the training scheme across different datasets or T2V/mi2v models. The table results only show the empirical choice of different probabilities on the T2V/mi2v tasks.
> >
> > It is also a bit confusing regarding the setting with t2v0.9&mi2v0.1, where a degraded performance was observed. Though the response has contemplated that it could be due to the fact that the model was converged, this explanation was not sufficient. Are there more insightful conclusions? Could the response also provide the experimental results for t2v0.5&mi2v0.5? It looks like the proposed method prefers more balanced distribution with t2v0.6&mi2v0.4 (ours).
> >
> > The difficult-to-easy training was not very theoretically motivated. How about the proposed strategy compared to the curriculum learning? The latter should also applies to the proposed task in this paper.

---

> > > ### Author Response · Authors · 2025-11-28
> > > **Additional Official Comment by Authors (Part 2/3)**
> > >
> > > `[W2] Confusion about t2v0.9&mi2v0.1 result.`
> > >
> > > In this experiment, we have trained all ablations 10k steps for a fair comparison. For t2v0.9&mi2v0.1 setting, the multi-ID task has been trained only 1k steps. However, we have only utilized a pret-rained t2v model as initialization. The model requires more training data to adapt to the reference image modality. To confirm this, we have additionally trained 10k steps for this setting. The results are shown below:
> > >
> > > |          Ablation          |   MUSIQ   |   CLIPIQA  |    QAlign   |  DOVER | CLIPI | DINOI |
> > > |:------------------------:|:---------:|:---------:|:---------:|:---------:|:----------------:|:---------------:|
> > > |              |      |      |  Text-to-video Generation    |      |             |            |
> > > |       t2v0.6&mi2v0.4 (ours)       |   56.422   |   0.375   |   4.510   |   0.780   |       -      |      -      |
> > > |       t2v0.9&mi2v0.1 (10k steps)      |   56.445   |   0.378   |   4.503   |   0.779   |       -      |      -      |
> > > |       t2v0.9&mi2v0.1 (20k steps)      |   56.952   |   0.385   |   4.498   |   0.782   |       -      |      -      |
> > > |              |      |      |  Multi-ID Image-guided Text-to-video Generation    |      |             |            |
> > > |       t2v0.6&mi2v0.4 (ours)       |   62.455   |   0.471   |   4.511   |   0.752   |       0.728      |     0.565      |
> > > |       t2v0.9&mi2v0.1 (10k steps)       |   58.007   |   0.421   |   3.839   |   0.688   |       0.691      |      0.529      |
> > > |       t2v0.9&mi2v0.1 (20k steps)       |   59.952   |   0.444   |   4.127   |   0.710   |       0.709      |      0.544      |
> > >
> > > As can be seen, additionally training 10k steps leads to a significant improvement in the multi-ID image-guided text-to-video generation task, further validating the hypothesis of model converge.
> > >
> > > Moreover, following your suggestion , we have added the comparison with t2v0.5&mi2v0.5 and t2v0.4&mi2v0.6 settings. Full comparisons are shown as follows:
> > >
> > > |          Ablation          |   MUSIQ   |   CLIPIQA  |    QAlign   |  DOVER | CLIPI | DINOI |
> > > |:------------------------:|:---------:|:---------:|:---------:|:---------:|:----------------:|:---------------:|
> > > |              |      |      |  Text-to-video Generation    |      |             |            |
> > > |       t2v0.4&mi2v0.6      |   55.278   |   0.359   |   4.302   |   0.755   |       -      |      -      |
> > > |       t2v0.5&mi2v0.5      |   56.321   |   0.376   |   4.500   |   0.779   |       -      |      -      |
> > > |       t2v0.6&mi2v0.4 (ours)       |   56.422   |   0.375   |   4.510   |   0.780   |       -      |      -      |
> > > |       t2v0.7&mi2v0.3      |   56.245   |   0.377   |   4.522   |   0.778   |       -      |      -      |
> > > |       t2v0.8&mi2v0.2      |   56.344   |   0.374   |   4.499   |   0.781   |       -      |      -      |
> > > |       t2v0.9&mi2v0.1      |   56.445   |   0.378   |   4.503   |   0.779   |       -      |      -      |
> > > |              |      |      |  Multi-ID Image-guided Text-to-video Generation    |      |             |            |
> > > |       t2v0.4&mi2v0.6      |   62.527   |   0.479   |   4.504   |   0.754   |       0.726      |     0.567      |
> > > |       t2v0.5&mi2v0.5      |   62.578   |   0.474   |   4.492   |   0.750   |       0.727      |     0.567      |
> > > |       t2v0.6&mi2v0.4 (ours)       |   62.455   |   0.471   |   4.511   |   0.752   |       0.728      |     0.565      |
> > > |       t2v0.7&mi2v0.3      |   62.271   |   0.465   |   4.488   |   0.755   |       0.727      |     0.563      |
> > > |       t2v0.8&mi2v0.2      |   62.344   |   0.469   |   4.434   |   0.749   |      0.725      |      0.560      |
> > > |       t2v0.9&mi2v0.1      |   58.007   |   0.421   |   3.839   |   0.688   |       0.691      |      0.529      |
> > >
> > > From the results, the result of t2v0.5&mi2v0.5 maintains a comparable quality to the settings used in our paper. However, when the probability of t2v task is further reduced to 0.4, the model's performance on the t2v task begins to decline.
> > >
> > > We believe that for the mi2v task, the model can use the input reference conditions to generate new details in a guided manner, which reduces the difficulty of this task compared with the t2v task. The model only needs a certain amount of training data in the early stage to adapt to the reference image modality. In general, if the number of training steps is sufficient, the training probability of different sub-tasks will not excessively affect the final quality of the model. Therefore, our exploration of the training probability is only to enable the model to converge faster across multiple sub-tasks.

---

> > > ### Author Response · Authors · 2025-11-28
> > > **Additional Official Comment by Authors (Part 3/3)**
> > >
> > > `[W3] Motivation of difficult-to-easy strategy.`
> > >
> > > We have already compared with curriculum learning setting in Tab.2 (named easy-to-difficult). It can be seen that our difficult-to-easy strategy achieves superior performance on both video quality and controlling ability. We believe that when simple tasks (such as text-guided video editing) are trained first, the model learns a shortcut--that is, instead of generating new details from text prompt, it turns to reducing the loss by copying and pasting pixel values from the reference video, which degrades its controlling ability of text modality. Therefore, when the model subsequently learns the text-to-video generation task, it becomes relatively difficult for the model to re-learn the paradigm of generating new details using text prompt, which hinders the formation of a unified framework.
> > >
> > > ---
> > > We hope the above statements can address your concerns. If you have any further questions, please let us know at any time.

---

> ### Author Response · Authors · 2025-11-26
> **Official Comment by Authors (Part 2/4)**
>
> `[W3] Computational complexity analysis.`
>
> Thank you for your valuable comment. We agree that the most consuming part would lie in the training phase. Meanwhile, we also indicate that this is precisely one of the side advantages of the method we proposed: UniMMVSR leverages visual reference conditions to generate corresponding high-frequency details in a guided manner, which reduces the difficulty of model fitting and prevents the model from generating random textures using text. The training phase contains 3 stages, and it takes around 1 day per stage.
>
> For the results shown in the main text, we use a pre-trained 10B text-to-video model as initialization to obtain better reference condition injection capability. Also, we claim that the proposed framework can be directly applied to a smaller backbone without modification. To confirm this, we replicate our framework using a pre-trained 1B t2v model as initialization and conduct a detailed comparison on the multi-ID image-guided task. The comparison is performed on a single H800 GPU for a 1080P video with 77 frames.  We keep other settings the same as the official repo for other baselines to maintain high-quality generation results of these methods. The results are shown as follows:
>
> |          Method         | MUSIQ | CLIP-IQA | QAlign | DOVER |   CLIP-I   |   DINO-I  | Params (B) | Time (s) | Memory (GB) |
> |:------------------------:|:---------:|:---------:|:---------:|:---------:|:---------:|:---------:|:---------:|:---------:|:---------:|
> |       Ours (10B)       |   62.248   |   0.465   | 4.428 | 0.745 | 0.726 | 0.566 | 10.0 | 1030.552 | 76.530 |
> |       Ours (1B)       |   61.278   |   0.458   | 4.382 | 0.733 | 0.719 | 0.562 | 1.0 | 237.931 | 46.258 |
> |       SeedVR       |   54.491   |   0.419   | 3.960 | 0.708 | 0.693 | 0.543 | 7.0 | 630.286 | 70.568 |
> |       STAR       |   58.810   |   0.449   | 4.282 | 0.763 | 0.696 | 0.546 | 2.0 | 555.347 | 62.220 |
> |       VEnhancer      |   60.656   |   0.469   | 4.149 | 0.707 | 0.671 | 0.533 | 2.0 | 542.346 | 60.554 |
>
> From these results, it can be seen that our 10B version achieves the optimal video quality and controlling ability with only a negligible increase in computational cost. Meanwhile, for our 1B version, although significantly reducing the model size leads to a decrease in controlling metrics, it still outperforms other baselines, demonstrating the effectiveness of the proposed framework.
>
> ---
> `[W4] Motivation of using multi-ID images.`
>
> To generate high-fidelity details, we leverage the high-frequency pattern of the referenced images/videos as guidance. In the bottom group of Fig.3, UniMMVSR successfully leverages the text information ``PERFECT DIARY'' provided by the left image. Also, in Fig.14, our method extracts the identity information from human face, clothes and the boat, which demonstrates the effectiveness of including visual references in the video super-resolution model.
>
> We agree that some references might not be useful during inference, and the model needs to automatically distinguish them. Therefore, we adopt a full attention mechanism: through a unified attention module, we use attention scores to automatically assign the importance of different reference conditions to the noise latent, thereby realizing more fine-grained injection.

---

> ### Author Response · Authors · 2025-11-26
> **Official Comment by Authors (Part 3/4)**
>
> `[W5] Advantages of UniMMVSR.`
>
> Thanks for your comment. The focus of UniMMVSR is to leverage multi-modal conditions to generate high-fidelity details. Therefore, the major improvements lie in the Text-guided Video Editing and Multi-ID Image-guided Text-to-video Generation tasks.
>
> As shown in Tab.1, our method outperforms other baselines in the controlling metrics by a large margin, which demonstrates the ability of our framework to generate high-fidelity details. For video quality, our method also achieves the best MUSIQ and QAlign metrics in the Multi-ID Image-guided Text-to-video Generation task. For Text-guided Video Editing task, since our method effectively injects referenced video information and maintains the consistency of non-editing area, our results are relatively close to the ``Ref Video''. We do not emphasize the superiority of UniMMVSR in the Text-to-video Generation task. We only include it to illustrate that our unified framework can also handle traditional super-resolution tasks.
>
> ---
> `[W6] Confusion about upsampler.`
>
> Thanks for your valuable comment. Upsampler denotes the sequential operations of VAE decoding, pixel upscaling via bilinear interpolation, and VAE encoding, which directly operates on low-resolution latent. We have added it in the caption of Fig.2. Following your suggestion, we have modified this part in Sec.3.2 to avoid confusion.

---

> ### Author Response · Authors · 2025-11-26
> **Official Comment by Authors (Part 4/4)**
>
> `[W7] Role of multi-ID images.`
>
> To generate high-fidelity details, we leverage the high-frequency pattern of the referenced images/videos as guidance. We agree that there is no relevance between the super-resolution output of text image and the referenced face image. However, we claim that this flexibility is one of the advantages of our framework. During inference, some references might not be useful, and the model needs to automatically distinguish them. Therefore, we adopt a full attention mechanism: through a unified attention module, we use attention scores to automatically assign the importance of different reference conditions to the noise latent, thereby realizing more fine-grained controll.
>
> ---
> `[W8] Reproducibility of UniMMVSR.`
>
> Thank you for your valuable comment. For the reproducibility, we will ensure the following points. Code: Our code and model will be made publicly available, including necessary scripts. Data: Detailed descriptions of our data processing are provided in Supp.A2. Experimental Setup: We have stated all experimental configurations, including hyperparameters, hardware specifications in Sec.4.1.1 of the main text. Model Architecture: The architecture details are described in method part. We will also replicate our model using public Wan2.1 framework in the next version.
>
> ---
> `[W9] Token-concat text, images and videos.`
>
> Thanks for your insightful comment. The UniMMVSR model is initialized from a pre-trained text-to-video model, with the text embedding injected by a 2D cross-attention module per layer. To faster convergence, we maintain the same architecture design for text modality as the base model to avoid time-consumed re-training. Following your suggestion, we have provided an additional ablation study on text-to-video generation and multi-ID image-guided text-to-video generation tasks to investigate the architecture design of the text modality. The results are shown as follows (CC denotes channel-concat, Cross denotes 2D cross-attention and TC denotes token-concat):
>
> |          Ablation          |   MUSIQ   |   CLIPIQA  |    QAlign   |  DOVER | CLIPI | DINOI |
> |:------------------------:|:---------:|:---------:|:---------:|:---------:|:----------------:|:---------------:|
> |              |      |      |   Text-to-video Generation   |      |             |            |
> |       Ours       |   56.422   |   0.375   |   4.510   |   0.780   |       -      |      -      |
> |       Cross text & TC others      |   56.120   |   0.379   |   4.422   |   0.782   |       -      |      -      |
> |       CC LR video & TC others      |   52.142   |   0.335   |   3.829   |   0.672   |       -      |      -      |
> |       Full TC      |   49.235   |   0.307   |   3.555   |   0.669   |       -      |      -      |
> |              |      |      |   Multi-ID Image-guided Text-to-video Generation   |      |             |            |
> |       Ours       |   62.455   |   0.471   |   4.511   |   0.752   |       0.728      |     0.565      |
> |       Cross text & TC others      |   62.399   |   0.488   |   4.582   |   0.751   |       0.730      |     0.562      |
> |       CC LR video & TC others      |   54.314   |   0.369   |   3.434   |   0.647   |      0.675      |      0.525      |
> |       Full TC      |   53.017   |   0.351   |   3.439   |   0.644   |       0.666      |      0.517      |
>
> For both tasks, token-concatenating low-resolution video leads to comparable performance as our UniMMVSR architecture. However, as can be seen in the CC LR video & TC others and Full TC lines, directly token-concatenating text embedding results in severe performance degradation on both video quality and controlling metrics. We hypothesize that token-concatenating text embedding requires intensive re-training, which is prohibitive on a video super-resolution task.
>
> ---
> We hope the above statements can address your concerns. If you have any further questions, please let us know at any time.

---

> ### Author Response · Authors · 2025-11-28
> **Additional Official Comment by Authors (Part 1/3)**
>
> We sincerely appreciate this reviewer's timely response and careful review of our rebuttal. Please check our detailed point-to-point responses to all the comments of this reviewer. We hope our responses can address this reviewer's concerns.
>
> ---
> `[W1] Generalization across different models and datasets.`
>
> Thanks for your valuable comment. We have already performed an additional experiment on a smaller backbone (1B t2v model) during the rebuttal period. The results on multi-ID image-guided text-to-video generation task are shown as follows:
>
> |          Method         | MUSIQ | CLIP-IQA | QAlign | DOVER |   CLIP-I   |   DINO-I  |
> |:------------------------:|:---------:|:---------:|:---------:|:---------:|:---------:|:---------:|
> |       Ours (10B)       |   62.248   |   0.465   | 4.428 | 0.745 | 0.726 | 0.566 |
> |       Ours (1B)       |   61.278   |   0.458   | 4.382 | 0.733 | 0.719 | 0.562 |
> |       SeedVR       |   54.491   |   0.419   | 3.960 | 0.708 | 0.693 | 0.543 |
> |       STAR       |   58.810   |   0.449   | 4.282 | 0.763 | 0.696 | 0.546 |
> |       VEnhancer      |   60.656   |   0.469   | 4.149 | 0.707 | 0.671 | 0.533 |
>
> From these results, it can be seen that our 10B version achieves the optimal video quality and controlling ability. Meanwhile, for our 1B version, although significantly reducing the model size leads to a decrease in controlling metric, it still outperforms other baselines, demonstrating the effectiveness of the proposed framework across different models. We will replicate our framework in the public Wan2.1 framework in the next version to further confirm this point.
>
> Moreover, we agree that the generality across different datasets should be verified. However, due to time constraints and the difficulty of data collection, the ablation experiments on the generalization of the dataset are still in preparation at present. We have included this part in the limitation section and will add this experiment in the next phase.

---

### Official Review · Reviewer_vfEy · 2025-10-28

**Soundness:** 3
**Presentation:** 3
**Contribution:** 3
**Rating:** 6
**Confidence:** 5

**Summary:**

The paper proposes a new model, UniMMVSR, that upscales AIGC videos while also respects hybrid conditions including low-res input, text, ID images and other videos. With multi-modal inputs, it is able to upscale a video up to 4K. It also shows a degradation pipeline based on SDEdit. It shows outstanding performance among multiple video generation tasks.

**Strengths:**

### Motivation
- UniMMVSR is designed for AIGC video super-resolution with a cascaded model after the base model. It shares the same latent space with the low-res videos.
- To obtain high-fidelity results, UniMMVSR uses multiple inputs as conditions. It unifies multi-modalities including texts, images and videos.

### Method
- Token-wise concatenation makes sense for reference input, given that they are not pixel-aligned with the low-res input video.
- Separated conditional RoPE also makes sense for reference tokens.
- To simulate AIGC artifacts, authors use SDEdit pipeline to "degrade" the input videos.
- Reference augmentation is used to better adopt references in different scenarios.

### Experimental results
- Qualitative results show UniMMVSR can preserve details from reference inputs in Fig. 3.
- UniMMVSR shows competitive qualitative results in three tasks on different no-reference metrics, especially in Multi-ID reference video generation.

### Writing
The paper is easy to follow and well-written.

**Weaknesses:**

### Motivation
- The title sounds a little misunderstanding to me. UniMMVSR is designed to super-resolve AIGC videos, but authors use the term "Cascaded Video Super-Resolution", which is more often used for multiple, cascaded networks for video super-resolution in my opinion. Maybe use "Reference-based Video Super-Resolution" or other terms would be better than "Cascaded Video Super-Resolution".

### Method
- Since most of the VAEs are lossy, will "decoding LR latent -> pixel upscaling -> VAE encode" further increase the information loss? (Sec 3.2, Low-resolution video via channel concatenation).
- The token-wise concatenation is not well documented. What strategies are used if the number of reference tokens are not the same, i.e. variable token length? Are there "NULL" tokens for padding during the training/inference?

### Experimental results
- 4K results have some limitations. (a) Too short, only 1s (or 21 frames). (b) Color shift (e.g., 41397180.mp4 in text-to-video generation. The first frame and the last frame look so different).
- The qualitative results are mainly compared on no-reference metrics such as MUSIQ, DOVER, etc, which are infamous for their bias towards their training data. Can authors show a user study as a supplement?

### Writing
- Typos: "our UniMMVSR model also need to" -> "our UniMMVSR model also needs to".

**Questions:**

- There is no very detailed information about the base model, such as model parameters, VAE parameters, etc. Is the base model a private model?

---

> ### Author Response · Authors · 2025-11-26
> **Official Comment by Authors (Part 1/3)**
>
> We sincerely appreciate this reviewer's insightful feedback and careful review of our manuscript. Please check our detailed point-to-point responses to all the comments of this reviewer. We hope our responses can address this reviewer's concerns.
>
> ---
> `[W1] Confusion about cascaded VSR.`
>
> Thanks for your advice. In the title, we use ``cascaded video super-resolution'' to refer to the video super-resolution model in the cascaded generation framework. Following your suggestion, we have modified the title and unified it throughout the main text to avoid confusion.
>
> ---
> `[W2] Information loss about upsampler.`
>
> Thanks for your insightful comment. We agree that the upsampling pipeline might leads to slight information loss. However, most VAE latent spaces do not support simple interpolation, which can cause significant structural distortion. Therefore, previous video super-resolution methods usually follow the ``vae decoding+pixel interpolation+vae encoding'' pipeline to align the spatial size. Following your suggestion, we further conduct an ablation study about noise augmentation technique effect in Fig.20. As can be seen, by setting noise augmentation timestep to 200, our method successfully preserves the global and local structures of the input video, demonstrating negligible information loss of the upsampling pipeline.

---

> ### Author Response · Authors · 2025-11-26
> **Official Comment by Authors (Part 2/3)**
>
> `[W3] Confusion about token-wise concatenation.`
>
> Thank you for pointing out. The referenced image/video are encoded to the latent space by the pre-trained VAE encoder. The reference latents are then concatenated with the noise latent in the temporal dimension and passed to the DiT model. To align the spatial size, we will pad zero pixel to the referenced image/video before VAE encoding. We have added this part in the main text following your suggestion.
>
> ---
> `[W4] Questions about 4K results.`
>
> Thanks for your suggestion. Due to data limit, we have collected 3.3w 4K videos with one-second length to demonstrate the resolution scaling ability of our cascaded framework. The color shift in case 41397180 might origin from slight model under-fitting. Following your suggestion, we have added a limitation section in Supp.A.7 to discuss about it. We will continue to expand the 4K dataset and extend the duration to 5 seconds. Also, we have updated our supplementary material to include more 4K videos for ``text-to-video generation'' part to demonstrate its scalability.

---

> ### Author Response · Authors · 2025-11-26
> **Official Comment by Authors (Part 3/3)**
>
> `[W5] User study.`
>
> Thanks for your valuable suggestion. For further comprehensive comparisons, we carried out a user study that evaluated the results of both text-to-video generation and multi-ID image-guided text-to-video generation tasks. We included all three baselines in this study, consisting of SeedVR, STAR and VEnhancer. We invite a total of 20 participants for this user study. Each volunteer was presented with a set of 10 randomly selected video triplets, which included an input video, the results obtained from the compared methods, and our result. For the 10 video triplets, each of the two aforementioned tasks accounts for half. For the text-to-video generation task, their role was to choose the visually superior enhanced video from the given options. For the multi-ID image-guided text-to-video generation task, their role was to choose the visually similar video compared with the referenced images. The results are shown as follows:
>
> |          Task          |   Ours  |   SeedVR  | STAR | Enhancer |
> |:------------------------:|:---------:|:---------:|:---------:|:---------:|
> |       Text-to-video Generation       |   0.68   |   0.08   | 0.16 | 0.08 |
> |       Multi-ID Image-guided Text-to-video generation       |   0.91   |   0.06   | 0.02 | 0.01 |
>
> From the results above, it can be seen that our method generates more natural details for AIGC input than other baselines.  For multi-ID image-guided text-to-video generation task, results reveal a clear preference of controll ability among the volunteers for our results over those produced by other methods.
>
> ---
> `[W6] Typo.`
>
> Thanks for pointing out. We have revised it.
>
> ---
> `[W7] Details about base model.`
>
> We have used a private model as our base model. The base model comprises a 3D VAE and a latent DiT backbone, with 1.4B and 10.0B parameters respectively. Following your suggestion, we have added this part in Supp.A.1. We will replicate our framework in the public Wan2.1 framework in the next version to improve transparency.
>
> ---
> We hope the above statements can address your concerns. If you have any further questions, please let us know at any time.

---

### Official Review · Reviewer_Sjsc · 2025-10-30

**Soundness:** 3
**Presentation:** 3
**Contribution:** 2
**Rating:** 4
**Confidence:** 4

**Summary:**

This paper proposed a method to unify generative video super resolution framework to include multi-modalities, including text, images and videos. It helps to synthesize more details and achieve high fidelity outputs for various conditional inputs. Specifically, it first upscale and channel concatenate the low-res video latents with high-res latents. Then it treats the multi-ID images and reference videos as visual tokens and token-wise concatenate them with previous high-res latents. Such tokens are performed separate 2D self-attention by themselves and finally jointly performed 3D self-attention together with high-res tokens. For PE, it applies separate RoPE for each reference videos and the target video. The author also proposed SDEdit degradation which adds k steps noise to latents and decode back into RGB space and then add normal degradation.

**Strengths:**

- introduced additional information, including text, reference images, videos during video super resolution process, which makes the high-res detailed generated with guidance.
- Difficult-to-easy training order makes the multiple task performance better.
- Reference augmentation makes the output more robust.

**Weaknesses:**

- limited novelty of super resolution and reference-based generation task since LR channel concatenation and reference token-wise concatenation is typical in each domain of research. Using separate PE for reference tokens is also used in other papers.

**Questions:**

- The goal of SDEdit degradation is to mimic the real generated low-res video fidelity distribution. It would be helpful to has discussion about how close is it and propose any metrics to measure it.
- For individual RoPE assignment for multiple reference tokens, i.e., n_i to n_i + k_i, how are they selected? Does the order of reference content (images/videos) matter? Would the PE with smaller i biased during training?
- The output quality of base model might differs. How does the super resolution model handle difference quality inputs?

---

> ### Author Response · Authors · 2025-11-26
> **Official Comment by Authors (Part 1/2)**
>
> We sincerely appreciate this reviewer's insightful feedback and careful review of our manuscript. Please check our detailed point-to-point responses to all the comments of this reviewer. We hope our responses can address this reviewer's concerns.
>
> ---
>
> `[W1] The architecture novelty of UniMMVSR.`
>
> Thanks for your valuable comment. We agree that channel-concat and token concat have been proposed in previous methods. However, we would like to clarify that we have not regarded these two technologies as the main novelty of this paper. Instead, the main contributions of this paper are as follows: We have identified that the fundamental reason why existing cascaded generation frameworks are hardly applicable to various controllable generation tasks lies in the randomness introduced during the sampling process of traditional super-resolution models. Therefore, we have tackled this challenge by introducing multi-modal reference conditions to the video super-resolution model, and introduced a new task--multimodal video super-resolution.
>
> Moreover, we have found that the reference distortion problems are usually observed in the base model output, which typically does not exist in traditional video super-resolution tasks. Therefore, we follow the principle ``the reference distortion is equivalent to the results obtained by base model using only text condition'' and introduce a new SDEdit degradation pipeline to effectively simulate this scenario.
>
> By constructing a unified multi-task framework, we have made an influential discovery in the academic community: high-quality data can be transferred across different sub-tasks. We believe this discovery will help reduce the difficulty of data collection for complex reference tasks.
>
> ---
> `[W2] Quantitative evaluation about sdedit degradation.`
>
> Thanks for your insightful comment. We have already provided the detailed degradation pipeline and some samples in Supp.A4. The core of the sdedit degradation is to mimic the high-frequency pattern of the input video using text-to-video base model, which is consistent with the scenario where the foundation model has poor injection of visual reference conditions. The diffuse steps are randomly selected from 5 to 15 to preserve the basic structure of the input video.
>
> ● Following your suggestion, we conducted additional quantitative experiments to measure the identity similarity using CLIP-I and DINO-I metrics. Specifically, we randomly select 100 videos from training data, and apply Mask2former to extract the reference images from the first frame of the input video. The metrics are computed by the first frame of the input and sdedit-degraded videos:
>
> |          Video Type          |   CLIP-I   |   DINO-I  |
> |:------------------------:|:---------:|:---------:|
> |       Input Video       |   0.751   |   0.607   |
> |       5-steps Degraded Video       |   0.702   |   0.545   |
> |       15-steps Degraded Video       |   0.671   |   0.522   |
>
> From these results, we observe that the results of 5 steps maintain a certain level of identity similarity, while the results of 15 steps significantly corrupt the high-frequency identity features. This and Supp.A4 provide supportive evidence for our assumption that the sdedit degradation effectively mimics the generated video distribution. Light-degraded and heavy-degraded results can be seen in Fig.10 and Fig.11 respectively. Benefiting from the coverage and sampling flexibility of the proposed degradation pipeline, our model can process input videos of varying quality.

---

> ### Author Response · Authors · 2025-11-26
> **Official Comment by Authors (Part 2/2)**
>
> `[W3] Quantitative evaluation about the reference RoPE interval and its order.`
>
> Thank you for raising this important point. We follow two rules to assign the rope intervals: first, ref tokens of different categories do not overlap to avoid confusion; second, we assign smaller intervals for ref tokens of more difficult tasks to accelerate convergence. Specifically, we assign 0 to 20 for noise tokens, 20 to 30 for ref images and 30 to 50 for ref videos. To verify this, as you suggested, we have compare with an ablation (20 to 40 for ref images and 40 to 50 for ref videos) as follows:
>
> |          Models          |   MUSIQ   |   CLIPIQA  |    QAlign   |  DOVER | CLIPI | DINOI | PSNR | SSIM | LPIPS |
> |:------------------------:|:---------:|:---------:|:---------:|:---------:|:----------------:|:---------------:|:-------------:|:----------------------:|:--------------------:|
> |              |      |      |      |  Text-guided Video Editing    |             |            |          |                  |           |
> |       Ours       |   53.245   |   0.344   |   4.305   |   0.597   |       -      |      -      |     31.556     |          0.713         |         0.282   |
> |       Ablation      |   53.242   |   0.345   |   4.310   |   0.595   |       -      |      -      |     31.555     |          0.715         |        0.281   |
> |              |      |      |      |  Multi-ID Image-guided Text-to-video Generation    |             |            |          |                  |           |
> |       Ours       |   62.248   |   0.465   |   4.428   |   0.745   |       0.726      |      0.566      |    -     |          -         |         -   |
> |       Ablation      |   62.210   |   0.462   |   4.412   |   0.744   |      0.720      |      0.561      |     -     |          -         |        -   |
>
> As can be seen, switching the rope intervals results in comparable video quality. For the ablation model, even when a relatively close interval is assigned, its control capability on simple tasks (video editing) does not improve significantly; however, its control capability on complex tasks (multi-image) decreases to a certain extent as a relatively distant interval is assigned (0.720 to 0.726 for CLIP-I and 0.561 to 0.566 for DINO-I).
>
> ---
> `[W4] Handle different quality input.`
>
> Thanks for your insightful comment. The motivation of our degradation pipeline is to flexibly handle base model output of different quality. As stated in the main text, we seperate the degradation scenarios into two types: lack of details and reference distortion. For lack of details, we utilize synthetic degradation factors such as gaussian blur, resize and compression to simulate. During training, we randomly sample the degradation intensity following the pipeline in RealBasicVSR, which can effectively handle model output whether it is clear or blurry.
>
> For reference augmentation, we propose a new degradation pipeline called SDEdit degradation, which corrupts the high-frequency pattern of the base model output in the latent space by adding gaussian noise using diffuse process. This strategy effectively lowers the high-frequency similarity of base model output to the reference images/videos. By setting appropriate threshold, our model learns to handle base model output with diverse similarity to the references.
>
> ---
> We hope the above statements can address your concerns. If you have any further questions, please let us know at any time.

---

> ### Author Response · Authors · 2025-11-28
> **Follow-up on Rebuttal Response for Paper 1451**
>
> Dear Reviewer Sjsc,
>
> We hope this message finds you well.
>
> As the discussion period is coming to a close, we would like to kindly ensure that our response and the **extensive new experiments** conducted based on your constructive suggestions have reached you.
>
> In our rebuttal, we have made significant updates to address your concerns:
>
> * **Clarification on Novelty (W1):** We clarified the distinction between our UniMMVSR and methods like FullDiT/FullDiT2. For the first time, we introduce **multi-modal reference conditions** to the video super-resolution model of the **cascaded generation framework**, which successfully extends the cascaded framework to various controllable generation tasks. Also, to solve the reference distortion problem, we introduce a new **SDEdit degradation pipeline** to synthesize the degraded patterns by a pre-trained text-to-video base model, which effectively simulates this scenario. Furthermore, we have made an influential discovery in the academic community: **high-quality data can be transferred across different sub-tasks**. We believe this discovery will help reduce the difficulty of data collection for complex reference tasks.
> * **Evaluation of sdedit degradation (Q1):** To quantitatively evaluate the effect of the proposed SDEdit degradation, we randomly select 100 samples from training data and evaluate the similarity with the reference images by **CLIP-I** and **DINO-I** scores. The results in Tab.4 provide supportive evidence for our assumption that the sdedit degradation effectively mimics the generated video distribution.
> * **Comparison of RoPE interval and order (Q2):** Following your suggestion, we have conducted an ablation study about the RoPE interval and order. As shown in the updated Tab.9, **switching the RoPE intervals results in comparable video quality**, which demonstrates the robustness of the proposed framework.
> * **Robustness of different quality input (Q3):** We have added a detailed explanation about the proposed degradation pipeline, which includes sdedit and synthetic degradations. Light-degraded and heavy-degraded results can be seen in Fig.10 and Fig.11 respectively. Benefiting from the coverage and sampling flexibility of the degradation pipeline, our model can process input videos of varying quality.
>
> Given the limited time remaining in the discussion period, we would be extremely grateful if you could verify whether our extensive new experiments and clarifications help resolve your initial concerns. **If you find that our response and these new results have successfully strengthened the paper and addressed your questions, we would highly appreciate it if you could consider raising your score to reflect these significant improvements.**
>
> Thank you again for your time and valuable input to improve our work.
>
> We remain available to answer any further questions.
>
> Best regards,
>
> The Authors of Submission 1451

---

### Author Response · Authors · 2025-11-26
**Global Reply**

Dear Reviewers,

We sincerely thank all reviewers for their thoughtful comments and constructive feedback.

We have carefully considered each point and provided clarifications and justifications accordingly. Detailed responses are included below. In addition, we have uploaded a **revised PDF** in which all modifications are **highlighted in blue**. The **supplementary material** is also updated to include more video results. Specifically, we have made the following updates:

* **Title**: Modified the title to avoid confusion.
* **Section 3.2**: Modified the definition of upsampler and method part.
* **Section 4.1**: Added the training time.
* **Appendix A.1**: Added some details about base model.
* **Appendix A.2**: Added details of Image-related Transforms.
* **Appendix A.4**: Added a quantitative comparison about SDEdit degradation Effect.
* **Appendix A.5.4**: Added a quantitative comparison about Text Token-concat Design.
* **Appendix A.5.6**: Modified the content of video results in the supplementary material.
* **Appendix A.5.7**: Added a quantitative comparison about the reference RoPE interval and its order.
* **Appendix A.5.8**: Added a qualitative comparison about the Noise Augmentation Effect.
* **Appendix A.5.9**: Added a qualitative comparison about the Robustness against Sub-task Probability.
* **Appendix A.5.10**: Added a Computational Complexity Analysis.
* **Appendix A.5.11**: Added a comparison with a Smaller Backbone.
* **Appendix A.5.12**: Added a User Study.
* **Appendix A.6**: Added some details about Test Set.
* **Appendix A.7**: Added Limitations.
* **Appendix A.8**: Added Ethics Statement.
* **Appendix A.9**: Added Reproducibility Statement.

We hope that our explanations adequately address all concerns. Please feel free to let us know if any further details or clarifications would be helpful.

Best regards,

Authors of Paper #1451

---

### Author Response · Authors · 2025-12-03
**Summary for AC**

Dear Area Chair,

Given the recent reversion of review scores, we understand the increased workload during this period. To assist your final assessment, we provide a summary of our **core contributions and value**, the **reviewer consensus on strengths**, our **key revisions**, and the **updated reviewer assessments**.

---
**1. Core Contributions and Value**

* **Multi-modal Video Super-resolution:** We are the first to identify why **existing cascaded generation frameworks are hardly applicable to various controllable generation tasks**, as traditional super-resolution models **introduce randomness** to the final output during the sampling process. Therefore, we have tackled this challenge by introducing multi-modal reference conditions to the video super-resolution model, and introduced a new task--**multi-modal video super-resolution**.
* **SDEdit Degradation:** We have found that **the reference distortion problems are usually observed in the base model output, which typically does not exist in traditional video super-resolution tasks**. Therefore, we introduce a new SDEdit degradation pipeline to effectively simulate this scenario.
* **Multi-task Framework:** By constructing a unified multi-task framework, we have made an influential discovery in the academic community: **high-quality data can be transferred across different sub-tasks**. We believe this discovery will help **reduce the difficulty of data collection for complex reference tasks**.

---
**2. Reviewer Consensus on Strengths**

We sincerely appreciate the positive assessments from all reviewers:
* **Reviewer Sjsc** highlighted our UniMMVSR framework, which **makes the high-res detailed generated with guidance**. Also, he praised that the proposed strategy **makes the multiple task performance better** and **the output more robust**.
* **Reviewer vfEy** found the paper **easy to follow and well-written**, and claimed that the proposed designs **make sense**.
* **Reviewer jTBQ** praised the work as **the first one that can handle multimodal conditions simultaneously for the video super-resolution task**. Also, he admitted that the work **provides some useful insights** and **the experimental demonstrations also look solid**.
* **Reviewer YT7R** commended that the results demonstrate the effectiveness in **guiding the content generation with appealing performance**.

---
**3. Key Revisions**

We strictly followed the reviewers' suggestions to improve the paper's quality and have incorporated substantial updates into the revised PDF (highlighted as blue). Major updates include:

* **Motivation of Architecture Design (Reviewer Sjsc, jTBQ, YT7R):** We have revised the method part to avoid confusion. Also, we have added ablation studies across different model components in Appendix A.5.4, A.5.7, A.5.9 and A.5.11, which demonstrates the effectiveness of the UniMMVSR framework.
* **Implementation Details (Reviewer Sjsc, vfEy, jTBQ, YT7R):** Following the suggestion, we have added all necessary details in Section 3.2, 4.1 of the main text and Appendix A.1, A.2, A.6.
* **Computational Complexity Analysis (Reviewer jTBQ, YT7R):** We have conducted the computational complexity analysis in Appendix A.5.10. We also replicated our framework in a smaller 1B backbone to demonstrate the effectiveness across different models.
* **Supplementary Experiments (Reviewer vfEy, jTBQ):** We have supplemented experiments in Appendix A.4, A.5.8 and A.5.12. We have also provided more video results in the supplementary material for visualization.

---
**4. Updated Reviewer Assessments**

Following the rebuttal phase, the status of the reviewers is as follows:
* **Reviewer Sjsc**: Although **he did not participate in the entire discussion**, we have fully responded to his questions. We also posted a comment to request him to participate in the discussion, but we didn't receive any reply.
* **Reviewer vfEy**: We provided complete responses to the seven questions he raised. Although he did not reply, **his initial score of 6** indicated his willingness to accept this paper.
* **Reviewer jTBQ**: We provided detailed responses and experiments for his initial nine questions. After the initial response, **he showed great interest** and initiated a second round of responses, raising three new questions. However, after we provided extensive and detailed experiments and responses to fully address his concerns (**which may potentially lead to a higher score in subsequent responses**), the leak occurred and unexpectedly interrupted the discussion.
* **Reviewer YT7R**: After we provided complete answers to all his questions, he indicated that **our responses successfully resolved all his concerns and raised his score to 6**.

---
We respectfully request the Area Chair to consider **UniMMVSR's core value**, our **key revisions**, and the **available reviewers' feedback** when drafting the meta-review.

Best regards,

The Authors of Submission 1451

---

### Meta-Review · Area_Chair_yJL6 · 2026-01-05

**Summary:**

While several reviewers recognized the value of the first unified multi-modal video super-resolution framework, the consensus remains split due to concerns regarding architectural novelty and the robustness of the evaluation metrics. The Area Chair (AC) has carefully reviewed the submission, the rebuttal, and the subsequent discussion. While the empirical results are promising and the authors made significant efforts during the rebuttal—including adding a user study and complexity analysis —the remaining concerns regarding technical contribution and the subjective nature of the performance metrics suggest the paper is not yet ready for publication. Authors are strongly encouraged to expand upon the preliminary user study provided in the rebuttal  and further justify the architectural design in a future revision

**Reviewer Concerns:**

A primary concern raised by multiple reviewers is the limited architectural novelty, with some noting that the core components—such as channel and token-wise concatenation for condition injection—are standard practices in the field rather than fundamental innovations.

Reviewers pointed out potential biases in the no-reference metrics (e.g., MUSIQ, DOVER) used to evaluate generative quality.

The authors provided a computational complexity analysis and a user study with 20 participants  in response to reviewer requests. However, a more rigorous validation (larger scale user studies or deeper analysis) is needed.

**Reviewer Scores:**

Reviewer YT7R: Increased their score to a 6 during the discussion phase. They explicitly stated that the additional details, experiments, and the method's "potential for practical applications" resolved their concerns.

Reviewer jTBQ: Expressed strong interest and engaged in a deep technical dialogue involving a second round of questions. However, the discussion was interrupted unexpectedly, and the reviewer did not explicitly comment on a score increase before the phase ended.

Reviewer Sjsc: Maintained a score of 4. Unlike the other reviewers, Sjsc did not participate in the discussion phase or reply to the authors' rebuttal, leaving their initial concerns regarding incremental novelty standing by default.

Reviewer vfEy: Did not engage in the post-rebuttal discussion, leaving their score at a 6 (Weak Accept).

---

### Decision · Program_Chairs · 2026-01-26

Reject